# HoxC5 and miR-615-3p target newly evolved genomic regions to repress *hTERT* and inhibit tumorigenesis

TingDong Yan[1], Wen Fong Ooi[2], Aditi Qamra [2,3], Alice Cheung[1], DongLiang Ma [4,5], Gopinath Meenakshi Sundaram [6], Chang Xu[1,7], Manjie Xing[1,2], LaiFong Poon[1], Jing Wang[1], Yan Ping Loh[7,8], Jess Hui Jie Ho[1], Joscelyn Jun Quan Ng[1], Muhammad Khairul Ramlee[1], Luay Aswad[7,8], Steve G. Rozen [1,9], Sujoy Ghosh[10], Frederic A. Bard[11], Prabha Sampath[1,6], Vinay Tergaonkar[11], James O.J. Davies [12], Jim R. Hughes [12], Eyleen Goh [3,4,5], Xuezhi Bi [13], Melissa Jane Fullwood[7,8,11], Patrick Tan[1,3,7,9,14] & Shang Li[1,3]

The repression of telomerase activity during cellular differentiation promotes replicative aging and functions as a physiological barrier for tumorigenesis in long-lived mammals, including humans. However, the underlying mechanisms remain largely unclear. Here we describe how miR-615-3p represses *hTERT* expression. *mir-615-3p* is located in an intron of the *HOXC5* gene, a member of the highly conserved homeobox family of transcription factors controlling embryogenesis and development. Unexpectedly, we found that HoxC5 also represses *hTERT* expression by disrupting the long-range interaction between *hTERT* promoter and its distal enhancer. The 3′UTR of *hTERT* and its upstream enhancer region are well conserved in long-lived primates. Both *mir-615-3p* and *HOXC5* are activated upon differentiation, which constitute a feed-forward loop that coordinates transcriptional and post-transcriptional repression of *hTERT* during cellular differentiation. Deregulation of *HOXC5* and *mir-615-3p* expression may contribute to the activation of *hTERT* in human cancers.

[1] Cancer and Stem Cell Biology Program, Duke-NUS Medical School, 8 College Road, Singapore 169857, Singapore. [2] Cancer Therapeutics and Stratified Oncology, Genome Institute of Singapore, 60 Biopolis Street, Singapore 138672, Singapore. [3] Department of Physiology, Yong Loo Lin School of Medicine, National University of Singapore, 2 Medical Drive, Singapore 117597, Singapore. [4] Neuroscience Academic Clinical Programme, Duke-NUS Medical School, 8 College Road, Singapore 169857, Singapore. [5] Department of Research, National Neuroscience Institute, 11 Jalan Tan Tock Seng, Singapore 308433, Singapore. [6] Institute of Medical Biology (IMB), 8A Biomedical Grove, Singapore 138648, Singapore. [7] Cancer Science Institute of Singapore, National University of Singapore, 14 Medical Drive, Singapore 117599, Singapore. [8] School of Biological Sciences, Nanyang Technological University, 60 Nanyang Drive, Singapore 637551, Singapore. [9] Duke-NUS Centre for Computational Biology, Duke-NUS Medical School, 8 College Road, Singapore 169857, Singapore. [10] Cardiovascular and Metabolic Disorders Program, Duke-NUS Medical School, 8 College Road, Singapore 169857, Singapore. [11] Institute of Molecular and Cell Biology (IMCB), Singapore 138673, Singapore. [12] Medical Research Council (MRC) Molecular Haematology Unit, Weatherall Institute of Molecular Medicine, Oxford University, Oxford OX3 9DS, UK. [13] Bioprocessing Technology Institute, 20 Biopolis Street, Singapore 138669, Singapore. [14] Cellular and Molecular Research, National Cancer Centre, 11 Hospital Drive, Singapore 169610, Singapore. Correspondence and requests for materials should be addressed to S.L. (email: shang.li@duke-nus.edu.sg)

The ends of human linear chromosome are capped by telomeres[1,2]. Telomeres are synthesized by telomerase that consists of two core subunits, the protein subunit, hTert and the RNA subunit, hTR/hTERC[3,4]. Although hTR/hTERC is widely expressed, hTert and consequently telomerase activity are hardly detectable in the majority of human adult somatic cells, except for some stem cells and germ cells[3–7]. As a result, telomeres in normal somatic cells shorten progressively during each cell division, thereby limiting cell proliferation capacity and functions as an important barrier to prevent cancer initiation[1,8–10].

Pluripotent stem cells express robust telomerase activity to support their continuous proliferation[11–13]. Limited telomerase expression in adult tissue stem/progenitor cells also prevents accelerated telomere shortening and supports stem cell self-renewal for tissue regeneration and repair throughout our lifespan[7,14]. Genetic mutations in telomere- and telomerase-associated genes can lead to various diseases, termed telomere syndromes or telomeropathies, which are characterized by accelerated telomere shortening, premature aging and increase risk for cancer[15,16]. These findings highlight the importance of telomere homeostasis in human health.

Upon the induction of cellular differentiation, hTERT is repressed and eventually silenced in the majority of normal human somatic cells[17,18]. The repression of hTERT expression during cellular differentiation promotes replicative aging and may be an adaptive response to an increased mutation load arising from the evolution of homeothermy in long-lived mammals[19]. In contrast, both TR and TERT are highly expressed in most somatic tissues of mice. The mechanism underlying such phenotypic divergence in regulation of TERT expression in human and mouse tissues remains unclear. Previous studies using transgenic mouse lines with bacterial artificial chromosomes have uncovered that the cis-acting regulatory elements in the hTERT genomic locus are pivotal in mediating its silencing during normal development[20–22]. However, the identity of the cis-regulatory elements and the trans-acting factors that are required to induce hTERT silencing during cellular differentiation remains unclear.

Telomerase upregulation is observed in >85% of human cancers[3–6]. Recent studies have shown that mutations in hTERT promoter are the most frequent non-coding mutations in specific subsets of human cancers[23–26]. These mutations not only increase hTERT mRNA expression in cancer cells, but also abolish hTERT silencing during stem cells differentiation[27]. Therefore, a failure to suppress hTERT expression during normal cellular differentiation may be hijacked by cancer cells to activate telomerase expression during cellular transformation as well.

Here we have identified roles for HoxC5 and miR-615-3p in the negative regulation of hTERT in cancer cells and during differentiation of pluripotent stem cells. We found that HOXC5 and mir-615-3p are suppressed in pluripotent stem cells, but activated specifically and robustly from the same locus during cellular differentiation. Our data suggest that HoxC5 and miR-615-3p repress hTERT via an upstream enhancer region and 3′UTR, respectively. While HoxC5 and miR-615-3p are very well-conserved between human and mouse (identity = 99.5% and 100% respectively), the TERT 3′UTR and upstream enhancer regions are conserved in long-lived mammals such as chimpanzee and macaque, but not in short-lived mammals such as mouse and rat. These results indicate that the differential regulation of TERT expression in human and mouse relies on the divergence of cis-regulatory genomic elements developed recently during the evolutionary process.

In addition, overexpression of HOXC5 and miR-615-3p in human cancer cells significantly inhibits hTERT expression and suppresses cancer cell growth both in vitro and in vivo. Analysis of RNA-Seq data set from 33 TCGA cancer types indicated that reduced HOXC5 expression contributes to the activation of hTERT in human cancers such as thymoma and testicular germ cell tumors. These results uncover a developmental-controlled regulatory circuit constitute of the HOXC5 locus that represses hTERT by targeting recently evolved genomic elements in human cells. Loss of HoxC5-mediated hTERT repression may be an alternative mechanism in the activation of hTERT expression in human cancers, especially for cancers derived from tissues, such as thymus and testis, which contain telomerase-positive progenitor cells/stem cells.

## Results

**Distinct regulatory functions of the hTERT 5′UTR and 3′UTR.** hTERT is upregulated in >85% of all human cancers, and higher expression of hTERT mRNA is associated with higher telomerase activity[5,28]. Similar results were observed in a panel of pluripotent human embryonic stem (ES) cell line (WA01) and cancer cell lines with high variability (Fig. 1a, b). We further analyzed the correlation between hTERT mRNA levels, measured by real-time RT-PCR, and telomerase activity, measured by telomeric repeat amplification protocol (TRAP), in 56 cell lines in the NCI-60 panel. Regression analysis demonstrated that telomerase activity was moderately correlated to hTERT mRNA levels ($r = 0.49$), and only about 23% ($r^2 = 0.23$) of the variation in telomerase activity could be explained by changes in hTERT mRNA levels (Supplementary Fig. 1a), suggesting the presence of other regulatory mechanisms influencing hTERT expression.

To address whether post-transcriptional regulation may play a role in the regulation of hTert expression, we cloned the hTERT 5′ UTR (58 bp) and 3′UTR (560 bp) individually or together at the respective ends of the Renilla luciferase reporter gene (hRluc) in psiCHECK2 (Supplementary Fig. 1b). The psiCHECK2 vector also contains a constitutively expressed Firefly luciferase gene (hFLuc + ), which is used for normalization (Supplementary Fig. 1b). We found that different cancer cell lines and also WA01 human ES cells transfected with psiCHECK2-5′UTR all displayed lower normalized Renilla luciferase activity than those transfected with the control plasmid, psiCHECK2 (Fig. 1c). In contrast, the effects of the hTERT 3′UTR on Renilla luciferase activity were variable and cell line-dependent (Fig. 1d). Similar cell-line-specific results were observed when the cells were transfected with psiCHECK2-5′ + 3′UTR (Fig. 1e). Interestingly, cells with relatively lower telomerase activity generally also displayed relatively lower levels of Renilla luciferase activity upon transfection with psiCHECK2-3′UTR or psiCHECK2-5′ + 3′ UTR, and vice versa (compare Fig. 1b, d, e). These results suggest that the hTERT 3′UTR plays a role in regulating the differential expression of hTert in different cell lines.

**miR-615-3p is a novel negative regulator of hTERT expression.** MicroRNAs often regulate gene expression via 3′UTRs[29]. To investigate whether miRNAs are involved in the regulation of hTERT, we engineered HeLa reporter cells stably integrated with psiCHECK2-5′ + 3′UTR (hTERT) and performed a screen using the Dharmacon miRIDIAN® miRNA hairpin inhibitor library, which targets 757 mature miRNAs (miRBase release 10.0, Supplementary Fig. 2a). We performed three independent screens and plotted the Z score distribution of the $\log_2$ ratio for Renilla/Firefly luciferase activity (Fig. 1f). We identified eight miRNA inhibitors with a Z score >2.7 in multiple screens (p value = 0.0034, one-sided test), and verified that they increased the Renilla/Firefly luciferase activity ratio in psiCHECK2-5′ + 3′ UTR (hTERT) reporter cells (Fig. 1g). In addition, two of the miRNA hairpin inhibitors, targeting miR-483-3p and

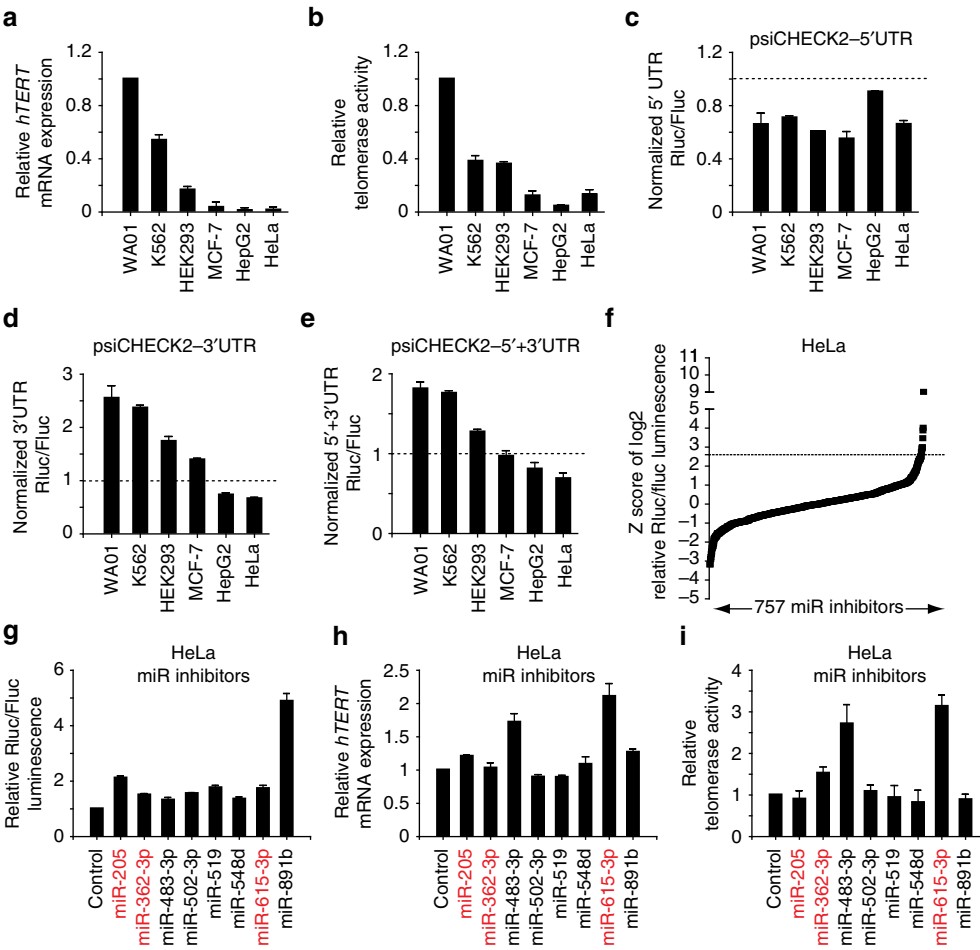

**Fig. 1** Identification of putative miRNAs targeting the 3′UTR of *hTERT*. **a** Expression of *hTERT* mRNA in different cell lines as quantified by real-time RT-PCR. The expression of *hTERT* mRNA in WA01 human ES cells is set as 1. **b** Telomerase activity in different cell lines as quantified by real-time TRAP. The telomerase activity in WA01 human ES cells is set as 1. **c–e** Relative Renilla/Firefly luciferase luminescence in different cells transiently transfected with psiCHECK2-5′UTR, psiCHECK2-3′UTR or psiCHECK2-5′ + 3′UTR vector comparing to the same cells transiently transfected with psiCHECK2 empty vector. **f** Z score distribution of the log$_2$ Renilla/Firefly luciferase luminescence ratio from three independent screens using a miRNA inhibitor library in HeLa cells stably expressing psiCHECK2-5′ + 3′UTR reporter. Candidates with Z score >2.7 were chosen for further analysis. The *p* value from one-sided test is 0.0034. **g–i** Relative Renilla/Firefly luciferase luminescence, *hTERT* mRNA expression and telomerase activity in HeLa cells transiently transfected with each of the eight candidate miRNA inhibitors. The miRNA candidates with reported differential expression in stem cells and terminal differentiated cells are highlighted in red

miR-615-3p, also dramatically increased endogenous *hTERT* mRNA levels and telomerase activity in HeLa cells (Fig. 1h, i). Neither of these miRNAs has been previously shown to regulate the expression of hTert. Interestingly, previous studies revealed that miR-615-3p level is low in pluripotent stem cells and increases dramatically upon differentiation[30,31], which is directly opposite to hTert expression in differentiated cells.

We compared the *hTERT* 3′UTR from different species[32] and found that it is highly conserved specifically in long-lived primates (Supplementary Fig. 2b), suggesting that the role of the *hTERT* 3′UTR in regulating *hTERT* expression was acquired late in genomic evolution. We used miRWalk 2.0[33] to identify potential miR-615-3p target site in the *hTERT* 3′UTR (Fig. 2a). We then mutated the predicted seed region (M1) as well as both seed region and an additional conserved site (M2) in the psiCHECK2-5′ + 3′UTR (*hTERT*) reporter construct (Fig. 2b). HeLa and RKO cells expressing psiCHECK2-5′ + 3′UTR-M1 (*hTERT*) displayed a ~30% increase in the Renilla/Firefly luciferase luminescence ratio compared to those expressing psiCHECK2-5′ + 3′UTR (*hTERT*) (Fig. 2c). Elimination of both the miRNA binding seed sequence and the additional conserved

site (psiCHECK2-5′ + 3′UTR-M2) further increased the Renilla/ Firefly luciferase luminescence ratio (Fig. 2c). Consistent with these results, overexpression of miR-615-3p further decreased Renilla/Firefly luciferase luminescence ratio in HeLa cells transfected with psiCHECK2-5′ + 3′UTR, but not psiCHECK2-5′ + 3′UTR-M2 (Supplementary Fig. 3a). These results suggest that miR-615-3p negatively regulates *hTERT* by targeting this region within the 3′UTR.

To further investigate the function of miR615-3p in regulating endogenous *hTERT* expression, we compared the expression profile of *hTERT* mRNA and miR-615-3p in NCI-60 cell lines. We found that HOP-92 and RXF393 cells displayed high levels of miR-615-3p but low levels of *hTERT* mRNA (Supplementary Fig. 3b). Further, transient transfection of the miR-615-3p hairpin inhibitor into these two cell lines led to a significant increase in *hTERT* mRNA expression and a corresponding increase in telomerase activity (Fig. 2d, e). In contrast, RPMI-8226, IGR-OV1 and HCC-2998 cells have low levels of miR-615-3p and high levels of *hTERT* mRNA (Supplementary Fig. 3b). We observed a significant reduction of *hTERT* mRNA levels and telomerase activity when miR-615-3p was overexpressed in these cell lines

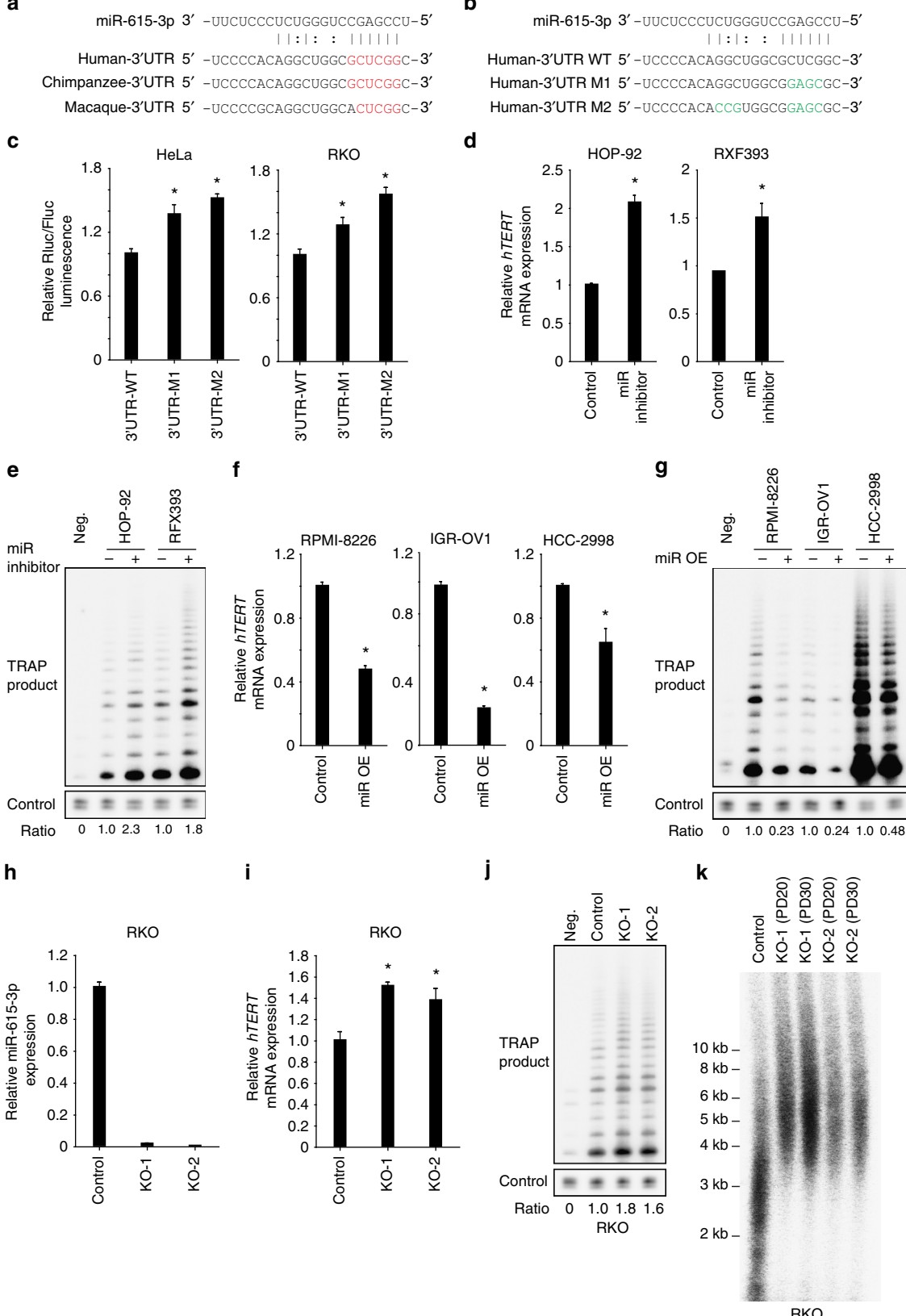

(Fig. 2f, g). Thus, miR-615-3p represses endogenous *hTERT* expression in various cancer cell lines.

To further confirm the function of miR-615-3p in the regulation of *hTERT* expression, we utilized the CRISPR/Cas9 system to introduce small deletion in the region of the *mir-615* hairpin that is crucial for its maturation. We isolated two clonal-derived RKO cell lines using two independent sgRNAs, RKO-KO-1 and RKO-KO-2 (Supplementary Fig. 4). As expected, RKO-KO-1 and RKO-KO-2 displayed loss of miR-615-3p expression (Fig. 2h), and increased *hTERT* mRNA levels and telomerase activity (Figs. 2i, j), relative to parental RKO cells. We also observed an increase in telomere length in RKO-KO-1 and RKO-KO-2 compared to parental RKO cells (Fig. 2k). We further utilized the CRISPR/Cas9 system to introduce small deletions or mutations into the endogenous *hTERT* 3′UTR. As shown in Supplementary Fig. 5a and 5b, clonal-derived RKO cells harboring biallelic deletions or mutations in the miR-615-3p-binding site displayed increased *hTERT* mRNA levels, but not in the cells harboring deletions outside the miR-615-3p-binding site. Taken together, these data suggest that miR-615-3p suppresses endogenous *hTERT* expression by targeting the 3′UTR of *hTERT*.

**HoxC5 inhibits the expression of *hTERT* in cancer cells.** The *mir-615* is located within intron 1 of the *HOXC5* gene (Fig. 3a), and the transcription of *HOXC5* is accompanied by the expression of miR-615[34]. Because *HOX* genes are important not only in development, but are also aberrantly expressed in many human malignancies[35], we investigated whether HoxC5 itself might influence the expression of *hTERT*. Indeed, we found that lentivirus-mediated overexpression of HoxC5, but not GFP, resulted in significantly reduced *hTERT* mRNA levels, reduced telomerase activity, and gradual telomere shortening in multiple cancer cell lines, including HeLa, PC-3, U-251 and BT549 cells (Fig. 3b–d). In contrast, we found that knockdown of the endogenous *HOXC5* gene in HeLa cells, using two independent lentiviral-expressed anti-*HOXC5* shRNAs, resulted in reduced *HOXC5* mRNA levels (Fig. 3e), increased *hTERT* mRNA levels (Fig. 3f), and telomere elongation (Fig. 3g). Inactivation of miR-615 expression does not affect the expression of endogenous *HOXC5* (Supplementary Fig. 5c, d). Together, our data suggest that miR-615-3p and HoxC5 form a feed-forward loop to negatively regulate *hTERT* mRNA expression, telomerase activity and telomere elongation.

**Activation of *mir-615-3p* and *HOXC5* upon differentiation.** Hox proteins transcriptionally regulate key target genes that are required for vital developmental processes[36,37]. Previous studies have shown that Hox proteins form complexes with Pbx and/or Meis - TALE (Three Amino acid Loop Extension) superfamily homeodomain proteins to acquire DNA-binding specificity[38,39]. There are four Pbx proteins and three Meis proteins expressed in humans. We investigated the expression patterns of *hTERT*, *mir-615-3p*, *HOXC5*, *PBX1–4* and *MEIS1–3* during the differentiation (Fig. 4a) of human ES cells WA01 and WA18 into neural progenitor cells in vitro[40,41]. As expected, the differentiation of ES cells into neural progenitors triggered rapid downregulation of *OCT-4* and *NANOG*, rapid induction of the neural progenitor marker *PAX6* (Fig. 4b and Supplementary Fig. 6a), and also the gradual downregulation of *hTERT* expression (Fig. 4c and Supplementary Fig. 6b). Importantly, we observed a dramatic increase in both miR-615-3p (Fig. 4d and Supplementary Fig. 6c) and *HOXC5* mRNA levels (Fig. 4e and Supplementary Fig. 6d), which plateaued around P2–P4. While the expression of *MEIS1* and *MEIS2* is also dramatically upregulated upon neural induction of ES cells, we only observed moderate increases in *PBX4* and *MEIS3* expression, and little changes in *PBX1*, *2*, and *3* expression (Fig. 4f and Supplementary Fig. 6e). These results indicate that *mir-615-3p*, *HOXC5*, *PBX4*, *MEIS1*, and *MEIS2* are dramatically upregulated upon differentiation of human ES cells into neural progenitors.

**The HoxC5 homeodomain is required to repress *hTERT*.** Previous studies have shown that the homeodomain (HD) and hexapeptide (HX) motifs within Hox proteins are important for their function[42,43]. The HX motif contains a short peptide, YPWM, which is important for Hox protein interactions with TALE superfamily homeodomain proteins in vitro, whereas the homeodomain (HD) is necessary for DNA binding[44]. To investigate whether these motifs are required for HoxC5-mediated repression of *hTERT*, we engineered HoxC5 mutants with disruptions of either the HX or HD motifs (Fig. 5a). We found that mutation of the HD domain, but not the HX domain, abrogated HoxC5-mediated repression of *hTERT* and telomerase activity in both HeLa cells (Fig. 5b, c) and PC-3 cells (Supplementary Fig. 7a, 7b). The mutations in the HX or HD motifs did not affect the expression of HoxC5 in HeLa (Fig. 5d) and PC-3 cells (Supplementary Fig. 7c). Thus, the DNA-binding activity of HoxC5 HD is required to repress *hTERT* expression.

**HoxC5 recruits Pbx4 and Meis3 to repress *hTERT*.** Hox proteins form complexes with Pbx and/or Meis - TALE (Three Amino acid Loop Extension) superfamily homeodomain proteins to acquire DNA-binding specificity[38,39]. To identify HoxC5-binding partners that are necessary for the repression of *hTERT*, we investigated which Pbx and Meis proteins can interact with HoxC5 in cancer cells.

We found that immunoprecipitation of V5-tagged HoxC5 resulted in specific co-immunoprecipitation of Flag-tagged Pbx4, but not of Pbx1, 2, and 3 (Fig. 5e). Consistent with these results, immunoprecipitation of Flag-tagged Pbx4, but not of Flag-tagged Pbx1, 2 or 3, resulted in efficient co-immunoprecipitation of V5-tagged HoxC5 (Supplementary Fig. 7d). Further, we performed an in vitro binding assay and found that recombinant 6xHis-tagged HoxC5 can efficiently pull down in vitro translated and [35]S-Methionine labeled Pbx4, but not Pbx1, 2 or 3 (Fig. 5f).

**Fig. 2** The miR-615-3p negatively regulates *hTERT* expression in cancer cells. **a** Schematic representation of the predicted binding site of miR-615-3p within *TERT* 3′UTR across different species. The potential seed region is highlighted in red. **b** Schematic representation of mutations introduced at the predicted miR-615-3pbinding site in psiCHECK2-5′ + 3′UTR reporter vector. The mutated bases are highlighted in green. **c** Relative Renilla/Firefly luciferase luminescence in HeLa and RKO cells transiently transfected with wild-type or mutant psiCHECK2-5′ + 3′UTR reporter vectors. **d**, **e** Relative *hTERT* mRNA expression and telomerase activity in HOP-92 and RXF393 cells transiently transfected with control hairpin or miR-615-3p inhibitor. The telomerase activity in cells transiently transfected with control is set as 1. **f**, **g** Relative *hTERT* mRNA expression and telomerase activity in RPMI-8226, IGR-OV1 and HCC-2998 cells transiently transfected with control vector or vector overexpressing *mir-615-3p*. **h** Loss of endogenous miR-615-3p expression in two independent CRISPR/Cas9-mediated *mir-615-3p* knockout RKO cell lines. **i–k** Relative *hTERT* mRNA expression, telomerase activity and telomere length in *mir-615-3p* knockout RKO cell lines. PD: population doubling. Significance was determined by *t* test. *$P < 0.05$. OE overexpression

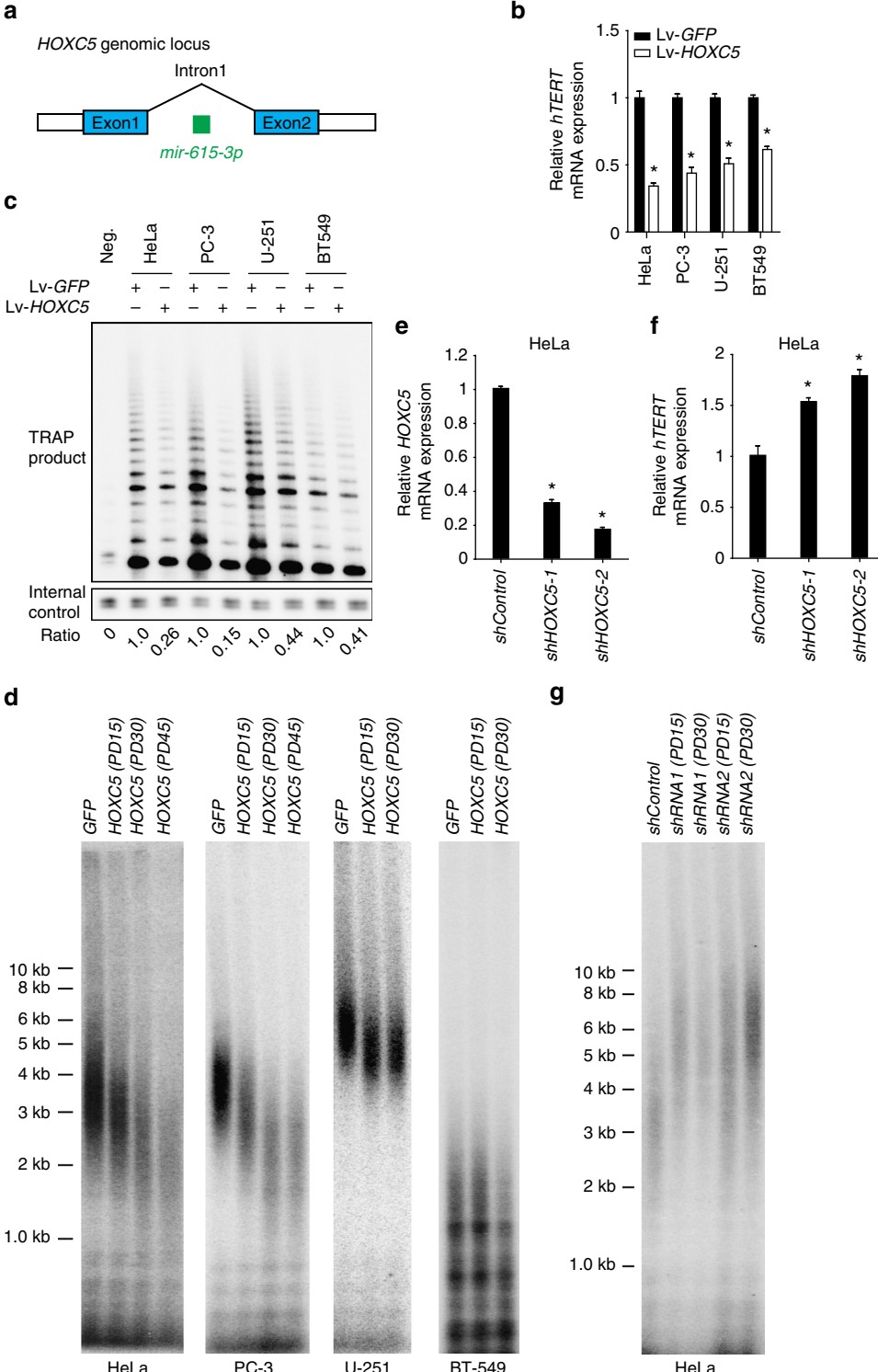

**Fig. 3** HoxC5 inhibits *hTERT* expression. **a** A schematic representation of *mir-615-3p* genomic localization in the intron of *HOXC5*. **b**, **c** Relative *hTERT* mRNA expression and telomerase activity in HeLa, PC-3, U251, or BT549 cells transduced with lentivirus overexpressing Flag-tagged *HOXC5*. The telomerase activity in cells transiently transfected with control GFP vector is set as (**1**). **d** Telomere length in HeLa, PC-3, U251, or BT549 cells transduced with lentivirus overexpressing *GFP* or Flag-tagged *HOXC5*. PD: population doubling. **e** The expression of endogenous *HOXC5* mRNA in HeLa cells expressing control shRNA or two independent anti-*HOXC5* shRNAs. **f** Relative *hTERT* mRNA expression in HeLa cells expressing control shRNA or two independent anti-*HOXC5* shRNAs. **g** Telomere length in HeLa cells transduced with lentivirus overexpressing control shRNA or two independent anti-*HOXC5* shRNAs. Significance was determined by *t* test. *$P < 0.05$. PD population doubling

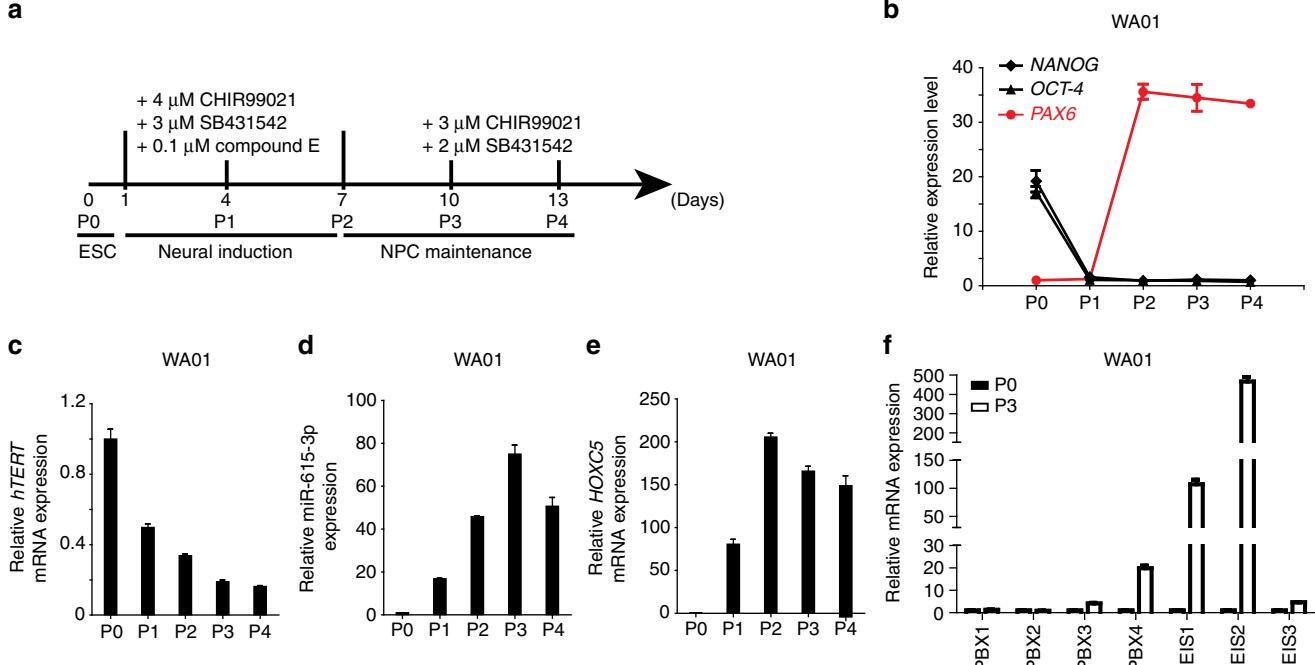

**Fig. 4** Expression of *hTERT*, miR-615-3p, *HOXC5*, *PBX1–4*, and *MEIS1–3* in WA01 human ES cells upon neural induction. **a** A schematic representation of the human ES cells neural differentiation process in monolayer culture. The details are described under "Methods". **b** The expression of pluripotency genes (*NANOG* and *OCT-4*) and neurodevelopmental gene *PAX6* during neural differentiation (passage 0–4) in WA01 embryonic stem cells were quantified by real-time RT-PCR. The expression of *hTERT* mRNA. **c** miR-615-3p (**d**) and *HOXC5* mRNA (**e**), during neural differentiation (passage 0–4) in WA01 cells was quantified by real-time RT-PCR as indicated. **f** The expression of *PBX1–4* and *MEIS1–3* in P0 and P3 of WA01 neural differentiation was quantified by real-time RT-PCR as indicated

Next we probed the interaction of HoxC5 and Meis proteins. We found that immunoprecipitation of Flag-tagged HoxC5 resulted in specific co-immunoprecipitation of V5-tagged Meis3, but not Meis1 and 2 (Fig. 5g). Consistent with these data, immunoprecipitation of V5-tagged Meis3, but not V5-tagged Meis1 and 2, resulted in efficient co-immunoprecipitation of Flag-tagged HoxC5 (Supplementary Fig. 7e).

Taken together, our results suggest that Pbx4 and Meis3 interact specifically with HoxC5. To investigate whether Pbx4 and Meis3 can synergize with HoxC5 to mediate the repression of *hTERT*, we transiently expressed GFP (control), HoxC5, Pbx4 and Meis3, alone or in combination, in HeLa cells and evaluated *hTERT* mRNA levels after 24 h. We found that transient overexpression of *HOXC5*, but not *GFP*, resulted in a reduction in *hTERT* mRNA levels (Fig. 5h). Although transient over-expression of *PBX4* and *MEIS3* alone did not significantly reduce *hTERT* mRNA levels, transient overexpression of *PBX4* or *MEIS3* in combination with *HOXC5* led to a synergistic reduction in *hTERT* mRNA levels (Fig. 5h). These results suggest that HoxC5 expression in HeLa cells is limiting, and the repression of *hTERT* by HoxC5 is enhanced by Pbx4 and Meis3.

**A HoxC5-Pbx4 complex recruits Class I HDACs.** Hox proteins can act as transcription activators or repressors in specific cellular contexts, depending on the recruitment of co-regulators. Hox–Pbx complexes have been shown to recruit HDACs for transcriptional repression[45]. In addition, the pan-HDAC inhibitor-Trichostatin A (TSA) can induce the expression of *hTERT*, suggesting that HDACs repress *hTERT* expression in certain cells[46,47]. To pinpoint which class of HDACs is necessary to repress *hTERT* in vitro, we treated PC-3 cells with inhibitors of class I HDACs (MS-275) or class II HDACs (MC1568). Similar to

TSA treatment, *hTERT* expression was activated by the inhibition of class I HDACs, but not of class II HDACs (Supplementary Fig. 8a).

To probe the interaction of HoxC5 and Pbx4 with class I HDACs, we transiently co-expressed V5-tagged *HOXC5* or V5-tagged *PBX4* with Flag-tagged *HDAC1*, *HDAC2* or *HDAC3* in HeLa cells (Supplementary Fig. 8b–d). We found that immuno-precipitation of V5-tagged HoxC5 resulted in co-immunoprecipitation of Flag-tagged HDAC3, but not HDAC1 and HDAC2 (Supplementary Fig. 8b). Slightly different from the results obtained with V5-tagged HoxC5, immunoprecipitation of V5-tagged PBX4 resulted in efficient co-immunoprecipitation of both HDAC1 and HDAC3, but not HDAC2 (Supplementary Fig. 8c and 8d). Together, these results suggest that a HoxC5:Pbx4 complex recruits HDAC1 and/or HDAC3 for repression of *hTERT*.

**HoxC5 represses *hTERT* by binding to upstream enhancer region.** We hypothesized that HoxC5 represses *hTERT* by binding to promoter-distal regulatory elements. However, the consensus binding motif for human HoxC5 is not known. Therefore, we profiled genome-wide HoxC5-binding sites by ChIP-Seq in PC-3 cells. We found that both endogenous HoxC5 and over-expressed Flag-tagged HoxC5 bind to a site −20 kb upstream of the *hTERT* transcriptional start site (TSS) in PC-3 cells (Fig. 6a). Consistent with the observed interaction between HoxC5 and Pbx4 (Fig. 5e), ChIP-Seq showed that overexpressed Flag-tagged Pbx4 also bound to the site −20 kb upstream of the *hTERT* TSS in PC-3 cells. This region is marked by H3K27ac and H3K4me1 in human ES cells and PC-3 cells, indicative of an active enhancer region, which is consistent with permissive expression of *hTERT* in these cell types. In contrast, IMR90 cells show depletion of

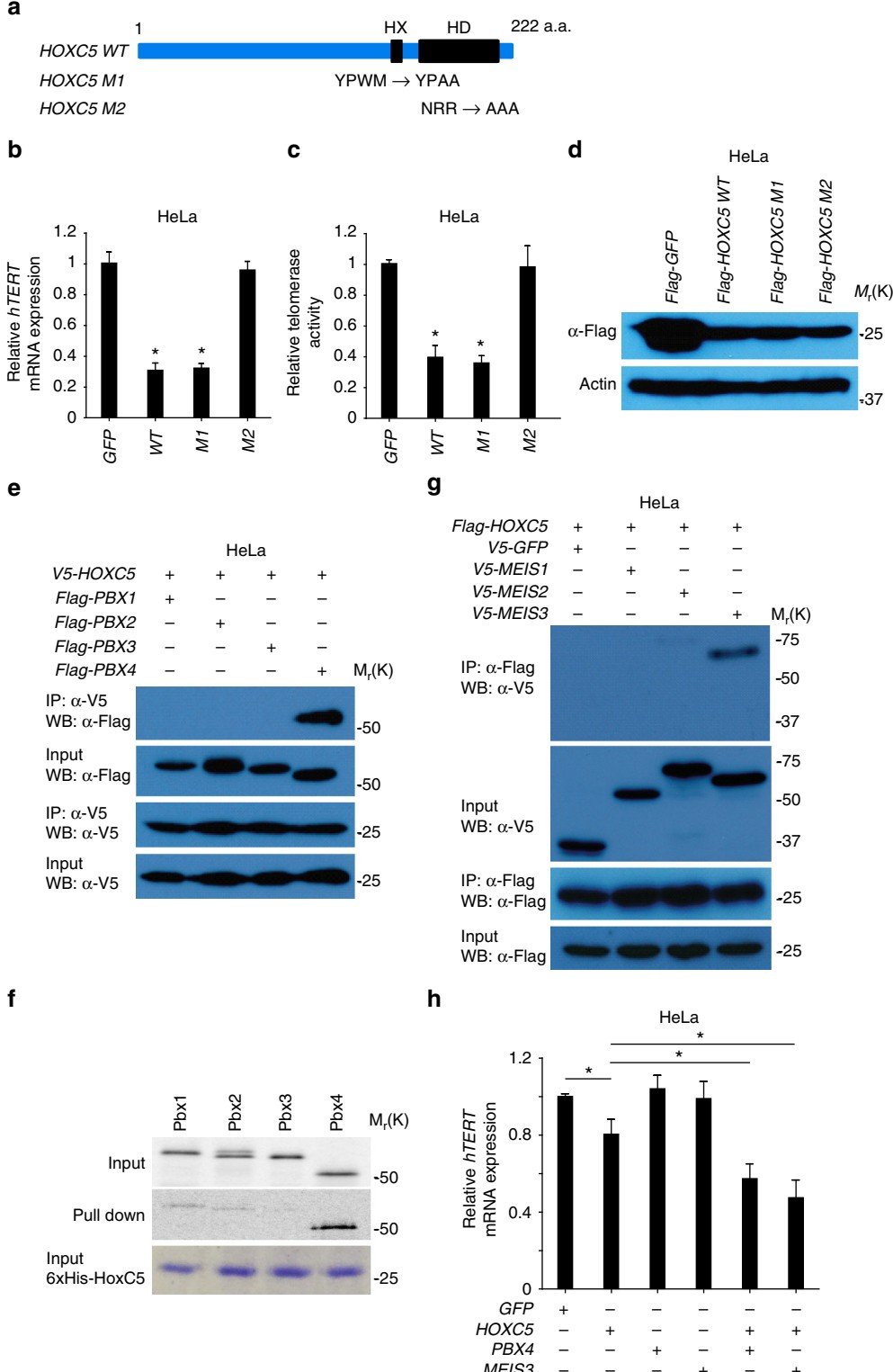

**Fig. 5** HoxC5 recruits Pbx4 and Meis3 to suppress *hTERT* expression in HeLa cells. **a** Schematic representation of *HOXC5* function domains with its HX and HD motifs highlighted in black. The mutations introduced in HX or HD motif are illustrated, respectively. **b**, **c** Relative *hTERT* mRNA expression and telomerase activity in HeLa cells transduced with lentivirus overexpressing Flag-tagged GFP, wild-type (WT), or mutants (M1 or M2) *HOXC5*. **d** Western blotting shows the expression of Flag-tagged GFP, wild-type (WT) and mutants (M1 or M2) HoxC5 expressed in HeLa cells. **e** Immunoprecipitation of V5-tagged HoxC5 resulted in specific co-immunoprecipitation of Flag-tagged Pbx4, but not Flag-tagged Pbx1, 2 or 3 in HeLa cells. **f** The purified recombinant 6xHis-tagged HoxC5 proteins from bacteria interact with the *in vitro* translated and [35]S-Methionine labeled Pbx4 proteins, but not Pbx1, 2, or 3. **g** Immunoprecipitation of Flag-tagged HoxC5 resulted in co-immunoprecipitation of V5-tagged Meis3, but not V5-tagged Meis1 or 2 in HeLa cells. **h**, Transient transfection of plasmids co-expressing *PBX4* or *MEIS3* with *HOXC5* results in synergistic suppression of *hTERT* mRNA expression in HeLa cells. Significance was determined by *t* test. *$P < 0.05$

H3K27ac at this region, and enrichment of H3K27me3 spanning the *hTERT* genomic locus, consistent with the silencing of *hTERT* in primary fibroblasts.

We used MEME to identify potential HoxC5-binding motifs within the HoxC5 ChIP-Seq peaks from endogenous HoxC5 in PC-3 cells[48], and uncovered a putative DNA-binding consensus (Supplementary Fig. 9a). A similar motif can be identified within the *hTERT* upstream enhancer region bound by HoxC5 (Supplementary Fig. 9b).

Next, we analyzed available 4C-seq data in A375 and BLM cells[49] to investigate the genomic loci interacting with the *hTERT*

promoter, and found that the 20 kb upstream enhancer region of *hTERT* interacts with the *hTERT* proximal promoter (Fig. 6a). In addition, overexpression of HoxC5 in PC-3 and A375 cells resulted in a corresponding depletion of RNA Pol II as well as histone marks associated with active gene transcription such as H3K4me1 and H3K27ac (Fig. 6b, c) at *hTERT* promoter and −20 kb enhancer, but not the −10 kb region. Similar results were observed upon neural differentiation of WA01 human ES cells (Fig. 6d).

To further address the importance of this upstream enhancer in the regulation of *hTERT* expression, we designed four specific

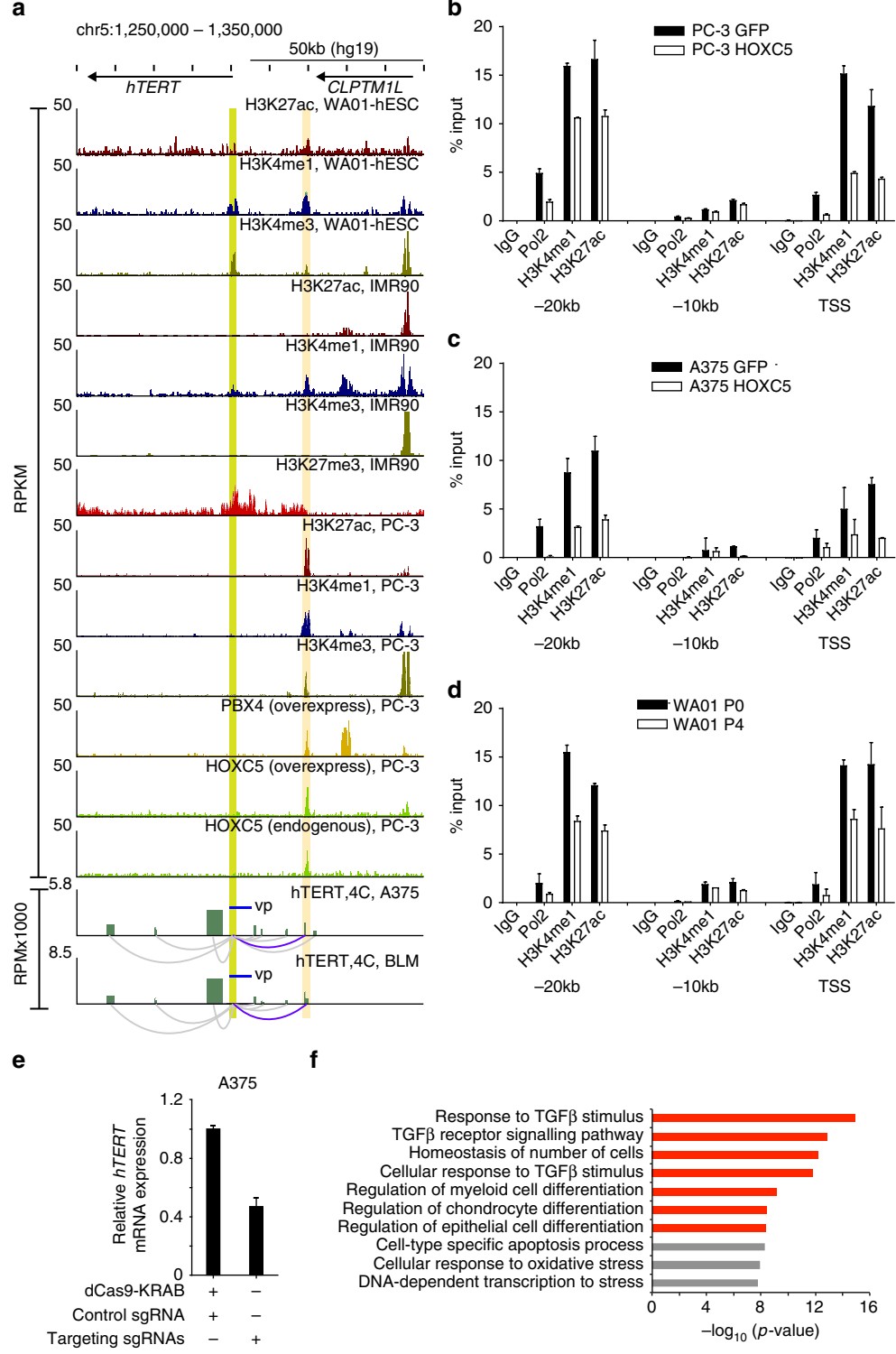

sgRNAs targeting the HoxC5-binding region (Supplementary Fig. 9b). Targeting of dCas9-KRAB fusion protein using these sgRNAs, but not control sgRNAs, to the *hTERT* upstream enhancer region also resulted in the suppression of *hTERT* expression in A375 cells (Fig. 6e) as well as depletion of RNA Pol II, H3K4me1 and H3K27ac at *hTERT* promoter and enhancer, but not at *hTERT* −10 kb region (Supplementary Fig. 9c).

The depletion of RNA Pol II, H3K4me1, and H3K27ac from *hTERT* promoter is likely due to the disruption of long-range interaction between *hTERT* promoter and the upstream enhancer by HoxC5, leading to the suppression of *hTERT* expression. As shown in Supplementary Fig. 10, overexpression of *HOXC5*, but not GFP, resulted in reduced interaction between *hTERT* promoter and its upstream enhancer specifically.

We further analyzed the sequence of this putative *hTERT* upstream enhancer region in different species and found that it is highly conserved in long-lived primates (Supplementary Fig. 11). Interestingly, we also found that human and mouse HoxC5 share extremely high identity (Supplementary Fig. 12a). In addition, overexpression of mouse *HOXC5* also resulted in suppression of *hTERT* expression in HeLa cells (Supplementary Fig. 12b), providing the possibility that mouse HoxC5 can mediate the silencing of *hTERT* by targeting the same region. In contrast, overexpression of miR-615-3p or *mHOXC5* does not affect the expression endogenous *mTERT* in three different mouse cell lines (Supplementary Fig. 12c–n).

***HOXC5* and miR-615-3p inhibit cancer cell growth**. We further analyzed genome-wide endogenous HoxC5-binding sites using Genomic Regions Enrichment of Annotations Tool (GREAT) and uncovered an over-representation of GO terms pertaining to cellular differentiation, including the TGF-β pathway[50,51] (Fig. 6f). Activation of genes in the TGF-β signaling cascade was confirmed by real-time RT-PCR in PC-3 cells overexpressing *HOXC5* and upon neural induction of normal human ES cells (Supplementary Fig. 13). In addition, telomerase is often aberrantly activated in cancer, and inhibition of *hTERT* expression suppresses cancer cell proliferation[52]. To study the potential role of miR-615-3p and HoxC5 in cancer cell proliferation, we transduced PC-3 cells with lentivirus expressing GFP, *HOXC5*, or *mir-615-3p* alone, or *HOXC5* and *mir-615-3p* in combination. As expected, we observed an increase in miR-615-3p levels in PC-3 cells transduced with lentivirus expressing *mir-615-3p* alone or co-expressing *HOXC5* and *mir-615-3p* (Supplementary Fig. 14a), as well as increased HoxC5 levels in PC-3 cells transduced with lentivirus expressing *HOXC5* alone or co-expressing *HOXC5* and *mir-615-3p* (Supplementary Fig. 14b). Consistent with our previous results, real-time RT-PCR revealed that overexpression

of *mir-615-3p* alone moderately reduced *hTERT* mRNA levels, overexpression of *HOXC5* alone resulted in >50% reduction in *hTERT* mRNA levels, and co-expression of *HOXC5* and *mir-615-3p* resulted in further repression of *hTERT* mRNA levels (Supplementary Fig. 14c).

We found that overexpression of *mir-615-3p* alone did not significantly affect the colony forming ability of PC-3 in vitro, although the colony size was smaller in PC-3 cells overexpressing *mir-615-3p* as compared to parental PC-3 cells (Fig. 7a, b). In contrast, overexpression of *HOXC5* significantly suppressed the ability of PC-3 cells to form colonies *in vitro*, and co-expression of HoxC5 and miR-615-3p further reduced the size of colonies without significantly affecting their number. Consistent with these results, overexpression of *HOXC5* in PC-3, BT549 and HeLa cells inhibited cancer cell proliferation in vitro (Supplementary Fig. 14d–f).

We established xenograft models to further validate the effects of miR-615-3p and HoxC5 on cancer cell growth in vivo. We injected PC-3 cells expressing GFP, *HOXC5*, or *mir-615-3p* alone, or co-expressing *HOXC5* and *mir-615-3p*, subcutaneously into immunocompromised NSG mice and followed the tumor growth for 5 weeks (Fig. 7c). At 5 weeks post-injection, tumors derived from PC-3 cells overexpressing *HOXC5* alone or in combination with *mir-615-3p* were significantly smaller than those over-expressing GFP alone (Figs. 7c, d, e). Overexpression of *HOXC5* in PC-3 also resulted in rapid telomere shortening (Fig. 3d) and induction of telomere-dysfunction induced foci (TIFs) (Supplementary Fig. 15), which may contribute to the inhibition of cancer cell proliferation.

**Deregulation of *HOXC5* expression in human cancers**. Telomerase upregulation is observed in >85% of all human cancers[3–7]. Although *hTERT* promoter mutations and genomic rearrangements have been shown to activate telomerase expression in a subset of human cancers[23–26,53], alternative mechanisms remain to be explored to understand the activation of telomerase in a vast majority of human cancers without known *hTERT* promoter mutations or genomic rearrangements. On the other hand, telomerase is highly expressed in several human adult tissues, such as thymus, testis and bone marrow[3–7]. Failure to suppress the expression of *hTERT* during cellular differentiation in these tissues due to genetic alterations may be hijacked by cancer cells to activate telomerase.

On the basis of the expression data from the HUMAN PROTEIN ATLAS, the expression of HOXC5 can be detected in multiple human tissues (Supplementary Fig. 16). Next, we wanted to examine the correlation of *hTERT* and *HOXC5* expression in human cancers. We examined the expression of *hTERT* and

---

**Fig. 6** HoxC5 represses *hTERT* expression through its binding to *hTERT* upstream enhancer region. **a** ChIP-Seq results indicate the binding of HoxC5 and Pbx4 at −20 kb upstream region (highlighted by yellow) of *hTERT* TSS (Highlighted by green) that show enrichment of H3K27ac and H3K4me1 specifically in telomerase-positive embryonic stem cells (WA01) and cancer cells (PC-3), but not in telomerase-negative primary fibroblast cells (IMR90). The interaction between *hTERT* promoter and the −20 kb enhancer region can be observed from 4C data done in A375 and BLM cells. *Cis*-interactions with FDR < 0.05 were selected for visualization (arcs). The blue arcs represent the interaction of hTERT promoter with HoxC5 and Pbx4-binding region (FDR = 6.3E-137). The gray color arcs are other significant interactions with *hTERT* promoter. **b** ChIP was performed against RNA polymerase II (Pol2), H3K4me1, and H3K27ac in PC-3 cells overexpressing control GFP or Flag-tagged *HOXC5* followed by qPCR with primers specific for *hTERT* TSS, −10 kb and −20 kb upstream enhancer regions as indicated. **c** ChIP was performed against RNA polymerase II (Pol2), H3K4me1, and H3K27ac in A375 cells overexpressing control GFP or Flag-tagged *HOXC5* followed by qPCR with primers specific for *hTERT* TSS, −10 kb and −20 kb upstream enhancer regions as indicated. **d** ChIP was performed against RNA polymerase II (Pol2), H3K4me1, and H3K27ac in WA01 cells before (P0) and after neural differentiation (P4) followed by qPCR with primers specific for *hTERT* TSS, −10 kb and −20 kb upstream enhancer regions as indicated. **e** The expression of endogenous *hTERT* mRNA in A375 cells co-expressing dCas9-KRAB with control sgRNAs or sgRNAs targeting the HoxC5-binding region at −20 kb upstream *hTERT* enhancer. **f**, The top ten biological processes associated with endogenous HoxC5-binding sites in PC-3 cells. The *p* value is based on the Binomial or Hypergeometric test. The pathways involved in cell differentiation are highlighted in red. vp view point

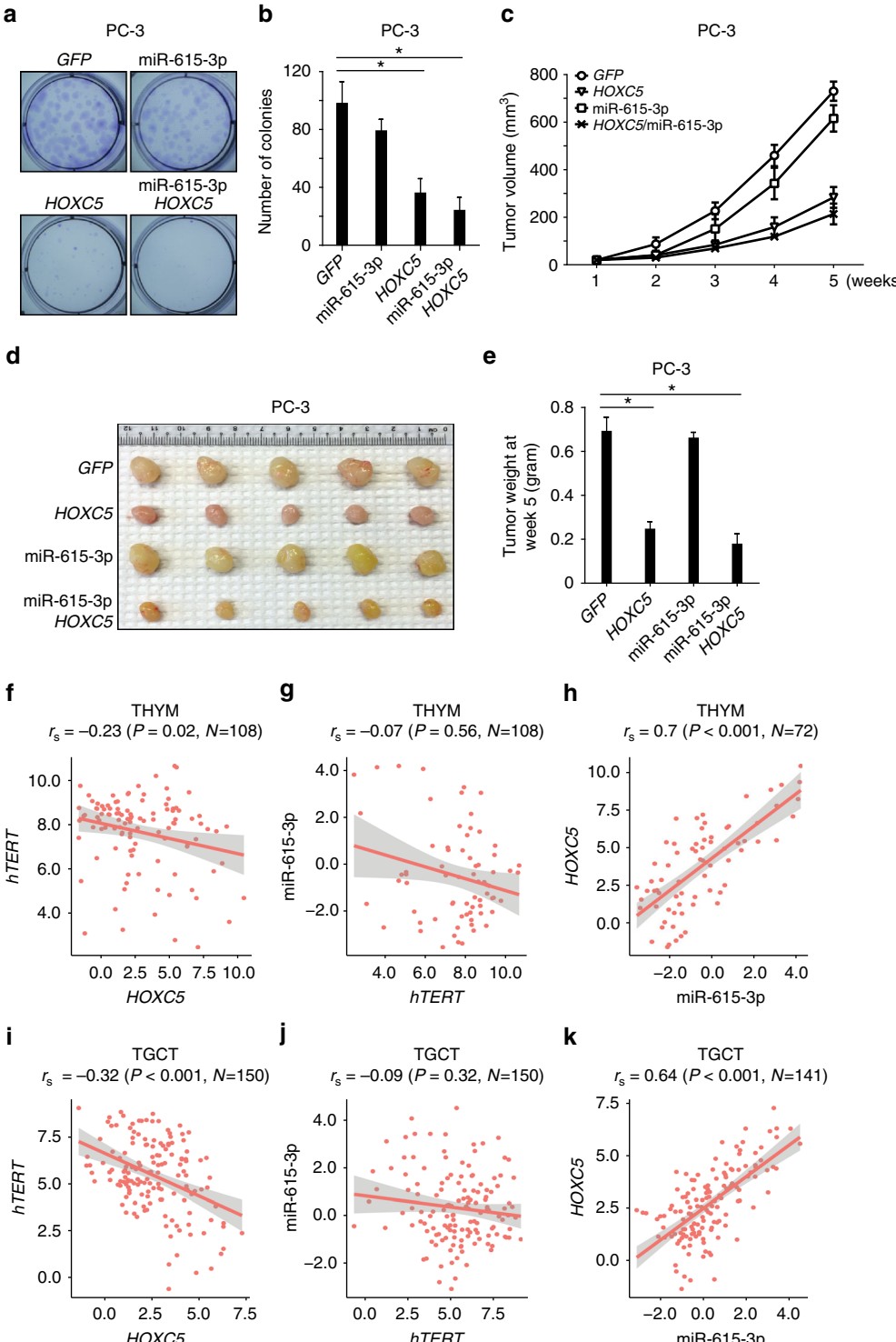

**Fig. 7** *HOXC5* suppresses *hTERT* expression and inhibits cancer cell growth. Representative images (**a**) and quantification (**b**) of in vitro colony formation assay in PC-3 cells expressing *GFP*, *mir-615-3p*, *HOXC5* or both *HOXC5* and *mir-615-3p*. Tumor growth curve (**c**) representing images (**d**), and tumor weight (**e**) showed growth of xenograft tumors in NSG mice injected with PC-3 cells expressing *GFP*, *mir-615-3p*, *HOXC5* or both *HOXC5* and *mir-615-3p* in vivo. Significance was determined by *t* test. *P < 0.05. Scatterplots showing Spearman correlation between *hTERT* and *HOXC5* (**f**, **i**); miR-615-3p and *hTERT* (**g**, **j**); *HOXC5* and miR-615-3p (**h**, **k**) in THYM (top) and TGCT (bottom) cancer samples. Red line indicates the linear fit with 95% confidence interval in shaded gray. All values are in RSEM log2 units. TGCT testicular germ cell tumor; THYM thymoma

*HOXC5* mRNA using RNA-Seq data in 33 cancer types from the TCGA consortium. As shown in Supplementary Fig. 17, the expression of *hTERT* and *HOXC5* in different human cancers has been wide spread. We observed significant correlation (Spearman's correlation, $P < 0.05$) in ~40% ($n = 13/33$) of the

cancer types between *HOXC5* and *hTERT* expression. However, only thymoma (THYM, Spearman $r = -0.23$, $p = 0.02$, Fig. 7f,) and testicular germ cell tumor (TGCT, Spearman $r = -0.32$, $p < 0.001$, Fig. 7i), which originated from telomerase-rich tissues, showed strongest inverse correlation between *HOXC5* and *hTERT*

expression. Interestingly, upon examining the expression pattern of *hTERT* and *HOXC5* across all 33 cancer types, THYM and TGCT have the highest median expression of *hTERT* (8.23 RSEM log2 units for THYM and 5.58 RSEM log2 units for TGCT) and a concordantly low expression of *HOXC5* (2.19 RSEM log2 units for THYM and 2.49 RSEM log2 units for TCGT). These results indicate the possibility that the reduced *HOXC5* expression may contribute to the activation of *hTERT* in cancers such as TGCT and THYM that rarely harbor *hTERT* promoter mutations[54,55].

Consistent with the observation that the expression of *HOXC5* and miR-615 is co-regulated[34], we also observed strong positive correlation between the expression of *HOXC5* and miR-615-3p (Fig. 7h–k) in THYM and TGCT. However, we did not observe significant negative correlation between the expression of miR-615-3p and *hTERT* (Fig. 7g–j). These results suggest that miR-615-3p only plays a fine-tuning role[56,57] in modulation of *hTERT* expression, while HoxC5 plays the key role in *hTERT* suppression.

## Discussion

Telomerase-directed telomere maintenance is essential during embryogenesis and development. In adult humans, telomerase activity is only detectable in germ cells and some stem cells, and is absent in the majority of normal somatic cells. The repression of telomerase in adult somatic cells limits their cell proliferation capacity and functions as a barrier to prevent tumorigenesis. However, the molecular mechanisms underlying the repression and eventual silencing of *hTERT* expression remain largely unknown. Here we show that the induction of *HOXC5* and miR-615-3p during stem cell differentiation represses *hTERT* via transcriptional and post-transcriptional pathways, respectively. These results are consistent with previous studies showing that *cis*-acting elements in the *hTERT* locus are necessary in mediating its silencing during normal development in transgenic mice[20,21].

In this study, we have shown that the *hTERT* 3′UTR plays an important role in the regulation of *hTERT* expression. Previous attempts to identify miRNAs that regulate hTERT expression have been limited by miRNA target-predicting programs[58]. Our unbiased genome-wide screen identified several miRNA candidates that target the 3′UTR of *hTERT*. Our data indicated that miR-615-3p negatively regulates *hTERT* expression by targeting *hTERT* 3′UTR. The expression of *mir-615-3p* likely functions as a micro-switch to further repress the expression of *hTERT*. Such a micro-switch likely sets a threshold for the expression of hTert protein until the *hTERT* mRNA reaches a significant level, thus preventing the leaky expression of genes such as hTert, after cellular differentiation[56,57]. Other potential targets of miR-615-3p were predicted using the microT-CDS algorithm (Supplementary Table 1). None of the potential targets seems to play a role in the regulation of hTERT expression directly.

The *mir-615* locus is located within the first intron of *HOXC5*, and it has a restricted phylogenetic distribution that arises in the ancestor of placental mammals. Interestingly, we found that HoxC5 also represses telomerase expression. Our data suggest that HoxC5 binds to an unanticipated upstream enhancer region of *hTERT*, therefore disrupting the long-range interaction between *hTERT* promoter and its upstream enhancer to repress *hTERT* expression.

Both miR-615-3p and Hox family proteins are well-conserved in mammals: human and mouse miR-615-3p have identical mature sequence and the identity between human and mouse HoxC5 is 99.5% (Supplementary Fig. 12a). Our results indicate that the 3′UTR and upstream enhancer region of *hTERT* are highly conserved in long-lived primates, but not in short-lived mammals (Supplementary Fig. 2b and Supplementary Fig. 11).

These results highlight that the divergence of *cis*-regulatory elements (*TERT* 3′UTR and upstream enhancer) developed during the recent evolutionary processes plays essential role in differential regulation of *TERT* expression in human and mouse. Consistent with this hypothesis, overexpression of mouse HoxC5 is able to suppress *hTERT* expression in HeLa cells (Supplementary Fig. 12b), while overexpression of miR-615-3p or mouse *HOXC5* in mouse cells does not affect the expression of *mTERT* (Supplementary Fig. 12c–n). It was shown previously that a 160-kb transgenic bacterial artificial chromosome (BAC) spanning the *hTERT* locus maintains its human species-specific silencing throughout mouse development[20–22]. This BAC contains the 3′UTR and upstream enhancer regions that we identified, and we propose that mouse miR-615-3p and HoxC5 repress *hTERT* via these elements.

Deregulation of *HOX* and *TALE* genes have been implicated in many human diseases, including cancer[35]. For example, strong expression of *HOXC5* can be detected in cells representing mature stages of lymphoid cell differentiation, while *HOXC5* expression could not be detected in leukemic cells (with activated *hTERT* expression) representing immature stages of lymphoid differentiation[59].

Our results indicate that both miR-615-3p and HoxC5, expressed from the same locus, repress *hTERT* expression during stem cell differentiation. Further, overexpression of miR-615-3p and HoxC5 suppress cancer cell growth both in vitro and in vivo in mouse xenograft model. Analysis of RNA-Seq data set from 33 TCGA cancer types suggests that reduced *HOXC5* expression contributes to the activation of *hTERT* in human cancers such as thymoma and testicular germ cell tumors. These results also suggested that the loss of *hTERT* repression by HoxC5 may be an alternative mechanism in activating *hTERT* expression in human cancers, especially for cancers (such as TGCT and THYM) derived from tissues with high telomerase activity. These results provide a novel mechanism for the activation of telomerase in human cancers in addition to known *hTERT* promoter mutations and translocations. Consistent with this hypothesis, *hTERT* promoter mutations are rarely identified in TGCT and THYM[54,55].

Thus, our study links the evolutionarily conserved Hox and TALE proteins to *hTERT* expression, which are all tightly coordinated with cellular differentiation and reprogramming. Understanding how miR-615-3p and HoxC5 mediate the cell-specific repression of *hTERT* and its implication in telomerase activation in human cancers can provide new avenues and targets for anti-cancer therapy.

## Methods

**Cell culture**. NCI-60 cell lines (ATCC) were cultured in RPMI medium with 10% fetal bovine serum and penicillin/streptomycin. Only 56 cell lines were used in this study, excluding MCF7ADRr, MDA-MB-435, MDN and OVCAR4. In addition, HEK293, HEK293T, HepG2, HeLa, and RKO cells were cultured in DMEM high glucose medium with 10% fetal bovine serum and penicillin/streptomycin. Feeder-independent human WA01 and WA18 embryonic stem cells from WiCell were grown on matrigel (BD Biosciences)-coated cell culture dishes using mTeSR1 culture medium (Stemcell Technologies)[41]. When the cells reached 80–90% confluence, they were passaged using Dispase (Stemcell Technologies), and split 1:6–1:12 onto new matrigel-coated cell culture dishes. No mycoplasma contamination was detected in cell lines used in this study.

**Molecular cloning**. The 5′- (58 bp) and 3′- (560 bp) UTRs of hTERT are cloned individually or together at the 5′ or 3′ ends of Renilla luciferase reporter gene (hRluc). The precursors of hsa-miR-615-3p plus upstream and downstream flanking sequences (total approximating 600 base pairs) were amplified from human genomic DNA isolated from WA01 cells. Flag-HOXC5, HOXC5 M1, HOXC5 M2, PBX1, PBX2, PBX3, PBX4, and V5-HOXC5, MEIS1, MEIS2, MEIS3, and GFP-miR-615-3p were constructed using pHR'CMVGFPIRESHygroWSin18 and HR'CMVGFPIRESPuro3WSin18 based lentivector[60].

**Antibodies and reagents.** Lipofectamine2000 (Thermo Scientific); goat anti-rabbit and sheep anti-mouse secondary antibodies conjugated to horseradish peroxidase (Jackson ImmunoResearch); antibodies for V5 (Thermo Scientific, #E10/V4RR), V5-HRP (Thermo Scientific, #R961-25), Flag (Sigma-Aldrich, #F1804), Flag (M2)-HRP (Sigma-Aldrich, #A8592), HOXC5 (Sigma-Aldrich, #HPA026794), PBX4 (Sigma-Aldrich, #HPA049859), PolII (Millipore, #05-623), Rabbit IgG (Millipore, #PP64B), H3K4Me1 (Abcam, #ab8895), H3K27ac (Abcam, #ab4729), Actin (Santa Cruz Biotechnology, #I-19); GAPDH (Cell Signaling, #2118) were used. EDTA-free complete protease inhibitor cocktail (Roche, #05056489001), Pan-HDAC inhibitor Trichostatin A (Sigma-Aldrich, #T1952), HDAC class I inhibitor MS-275 (Sigma-Aldrich, #EPS002), HDAC class II inhibitor MC1568 (Sigma-Aldrich, #M1824) were dissolved in dimethyl sulphoxide (DMSO).

**Generation of HeLa reporter cell line.** To engineer HeLa psiCHECK2-5'+3'UTR (hTERT) reporter cell line, the 5'- (58 bp) and 3'- (560 bp) UTRs of hTERT were cloned together at the 5' and 3' ends of Renilla luciferase reporter gene (hRluc) in psiCHECK2 vector using the available restriction enzyme sites. The puromycin drug selection marker was inserted in the Nhe I restriction enzyme site in psiCHECK2 plasmid. The plasmid was transfected into HeLa cells using lipofatemin2000. The cells were maintained in DMEM high glucose medium with 10% fetal bovine serum and penicillin/streptomycin with 0.5 µg/ml puromycin for two weeks, puromycin-resistant colonies were isolated and screened for Renilla and Firefly luciferase activity.

**Luciferase reporter assay.** For luciferase assay, 10,000 HeLa cells were seeded in 24-well plates. About 24 hr after seeding, cells were transiently transfected with 100 ng psiCHECK2, psiCHECK2-3'UTR, psiCHECK2-5'UTR, psiCHECK2-5' + 3'UTR (TERT) or psiCHECK2-3'UTR mutants using Lipofectamine 2000. About 16-24 hr after the transfection, cells were lysed in 1x Glo Lysis buffer (Promega) and luciferase activity was measured with the luciferase assay reagent (Promega) according to manufacturer's instructions. Luciferase activities were determined using a multimode microplate reader (Infinite 200 PRO,Tecan). Three independent experiments were performed for each group.

**Genome-wide screen to identify miRNAs that target hTERT.** For the genome-wide screen, the miRNA inhibitor library targeting 757 human miRNA inhibitors was printed on four 384-well plates (Grenier) with negative controls (miRNA inhibitor control) and positive controls (Firefly Luciferase siRNA and Renilla luciferase siRNA) in the designated wells. Plates were printed with miRNA inhibiter (2.5 µl of 500 nM miRNA inhibiter per well) and stored at -30 °C before use. For reverse transfection, a master mix of 0.02 µl of lipofectamine2000 and 4.98 µl of Opti-MEM (Invitrogen) was added to each well of the miRNA inhibitor plates and incubated for 20 min. Subsequently, 3000 HeLa reporter cells in DMEM medium were seeded in each well. Reagents and cells were dispensed onto the plate using a multidrop combi reagent dispenser (Thermo Scientific). Three days post transfection, Dual-Glo luciferase assay reagents (Promega) were used to measure the Firefly luciferase and Renilla luciferase activity according to the manufacturer's instructions. The ratio of Renilla/Firefly luminescence for each well was calculated.

**RNA isolation and quantitative RT-PCR.** For quantification of HOXC5, hTERT, PBX1–4 and MEIS1–3 mRNA expression, total RNAs were isolated using NucleoSpin RNA kit (Macherey-Nagel). About 300 ng of RNA from each sample was used for real-time RT-PCR using KAPA SYBR FAST Bio-Rad iCycler (Kapa Biosystems) and CFX96 Touch Real-Time PCR Detection System (Bio-Rad). GAPDH was used as internal control for normalization. For quantification of hTERT expression, the Taqman probe against TERT (Hs00972656_m1 FAM, Applied Biosystems) and GAPDH (Hs02758991_g1 FAM, Applied Biosystems) were used for real-time RT-PCR. For quantification of miR-615-3p expression, total RNAs were isolated using Trizol (Thermo Scientific). Reverse transcription of miR-615-3p and RNU6B (for normalization) was conducted using the TaqMan® MicroRNA reverse transcription kit (Thermo Scientific) followed by PCR using TaqMan® Fast Universal PCR Master Mix (2×), according to the manufacturer's instructions. U6 small nuclear RNAs were used as an internal control for normalization.

**Real-time TRAP assay.** Real-time TRAP assay was performed as previously described[61]. Briefly, the samples were spin down at 17,100 xg RCF for 20 min at 4 °C in a microcentrifuge. TRAP reaction was carried out in 1× TRAP buffer (20 mM Tris-HCl pH8.3, 1.5 mM MgCl₂, 63 mM KCl, 0.05% Tween 20 (Roche), 1 mM EGTA, 0.1 mg/ml BSA), 50 µM dNTPs, and 8 ng/µl TS Primer (5′-AATCCGTC-GAGCGAGAGTT-3′) in a 10-µl total volume for 30 min at 30 °C, and the reaction was stopped by heating at 94 °C for 1 min. The PCR reaction mixture contained 1X TRAP buffer, 4 ng/µl ACX primer 5′-GCGCGGCTTA CCCTTACCCTTACCC-TAACC-3′, 15% glycerol, 1:2000 SYBR Green from Invitrogen, 0.08 U/µl Taq polymerase and was initiated at 94 °C for 2 min, followed by 45 cycles of 95 °C for 5 s, 50 °C for 6 s, and 72 °C for 10 s. The threshold cycle values (Ct) were determined from semi log amplification plots. All samples were run in triplicates.

**Genomic DNA extraction and Southern blotting.** Genomic DNA was extracted using Gentra Puregene Genomic DNA Purification Kit (Qiagen). For telomere length measurement, the genomic DNA was digested with HphI and MnlI at 37°C for 16 hr. The DNA blot was hybridized with ³²P-labeled (TTAGGG)₆ oligonucleotides as previously described[41,62].

**CRISPR.** The miR-615-3p hairpin region was subjected to potential sgRNA target search using the online software created by Feng Zhang's group (http://crispr.mit.edu). The very top hits (sgRNA#1: 5′-CTTATTGTTCGGTCCGAGCCTGG-3′ and sgRNA#2: 5′-CACCCTCGAGATCCGAGCACCGG=3′) were chosen and used for the experiments hence described[63]. The isolation of single cell-derived clones with insertion and deletion is described previously[64]. CRISPR sgRNA used for engineer of hTERT 3′UTR truncation and knockin mutants is 5′-TGCCGTCTTCACTTCCCC-CACAGG-3′. The single-stranded oligo (5′-CGGCTGAAGGCTGAGTGTCCGGCT-GAGGCCTGAGCGAGTGTCCAGCCAAGGGCTGAGTGTCCAGCA-CACCTGCCGTCTTCACTTCCCCACTCCGTGGCCGGAGCCCTCCACCC-CAGGGCCAGCTTTTCCTCACCAGGAGCCCGGCTTCCACTCCCCACA-TAGGAATAGTCCATCCCCAGATTCGCCATTGTTCACC-3′) is used as template for knockin mutation of miR-615-3p binding site in the hTERT 3′UTR. CRISPR sgRNA used for targeting dCas9-KRAB to HoxC5-binding region are as follows: (1) 5′-GGTGTCAGATTACTAGGGCGGGG-3′; (2) 5′-CCCTAGTAATCTGA-CACCGCAGG-3′; (3) 5′-ACAATGGATCTTACCCACACAGG-3′ and (4) 5′-TGTGCTTAGCTGCTGCCAACAGG-3′.

**Western blotting.** Cell lysates were electrophoresed on a 10 or 12% SDS-PAGE gel and transferred onto polyvinylidene difluoride membranes, which were then blocked with 5% BSA in TBST [50 mmol/L Tris-HCl (pH 7.4), 150 mmol/l NaCl, and 0.1% Tween 20] for 1 hr, incubated with various primary antibodies overnight and detected with either anti-rabbit or anti-mouse secondary antibodies for 1 hr using SuperSignal West Dura Extended Duration Substrate (Thermo Scientific). The blots were re-probed with anti-actin or anti-GAPDH antibody for protein loading control. Primary antibodies used in the study are: anti-HOXC5 antibody (1:1000, Sigma, #HPA026794), anti-Flag (M2) antibody (1:5000, Sigma-Aldrich, #F1804) and anti–V5 antibody (1:5000, Cell Signaling, #13202). The blots were re-probed with anti-Actin (1:5000, Santa Cruz Biotechnology, #sc1616) or anti-GAPDH antibody (1:1000, Cell Signaling, #2118) for protein loading control. The uncropped western blot scans were shown in Supplementary Fig. 18.

**ChIP-Seq and analysis.** Cells were fixed with 1% formaldehyde for 10 min at room temperature, and stopped by adding glycine to a final concentration of 0.2 M. Cells were washed with ice cold TBSE (20 mM Tris-HCl, 150 mM NaCl, 1 mM EDTA) for three times. Chromatin were then extracted and sonicated to ~500 bp. Input DNA was precleared with protein G Dynabeads (Life technologies), and followed by chromatin immunoprecipitation with Hoxc5 (HPA026794, Sigma) and Pbx4 (HPA049859, Sigma) at 4 °C overnight. ChIP beads were washed six times at room temperature. At least 10 ng ChIP DNA were used to prepare libraries with NEB-Next ChIP-Seq library prep reagent set (New England Biolabs), and multiplexed (New England Biolabs). Each library was sequenced on Hiseq2000 (Illumina) to an average depth of 20-30 million reads with single-end 101 bp option[65].

Sequencing reads from ChIP-Seq were mapped against the human reference genome (hg19) using Burrows-Wheeler Aligner (BWA-MEM, version 0.7.0)[66]. High-quality mapped reads (MAPQ ≥10) reads were used for downstream analyses. Reads deemed as PCR duplication were removed using SAMtools. Binding peaks were detected using MACS2 with Q-value ≤ 0.05. Sequencing coverage was computed using MEDIPS[67] with a 50 bp window size and read length extension to 200 bp. DNA-binding motif was detected using MEME with the following parameters: -mod zoops -nmotifs 6 -minw 6 -maxw 50 –revcomp. Particularly, DNA sequences of top 500 binding sites of one transcription factor were used as input.

To investigate the pathways that are associated with endogenous HoxC5 bindings in PC-3, the binding sites were used in GREAT (http://bejerano.stanford.edu/).

Raw sequencing data in the format of.fastq and the processed data generated has been uploaded to GEO. Histone profiles of WA01 and IMR90 were obtained from Broad and REMS/UCSD, respectively through UCSC genome browser[68]. Mapped reads of PC-3 histone profiles (ENCFF537RRY, ENCFF297RXW, ENCFF122EOV, ENCFF945UYG) were downloaded from encodeproject.org.

**3C-qPCR materials and methods.** 3C-qPCR assays were performed as previously described[69] with slight modifications. Briefly, the assays were performed on PC3 cells overexpressed with HOXC5 or control GFP. $1 \times 10^7$ nuclei were cross-linked with 1% formaldehyde, digested using DpnII and ligated as described[69]. DNA was then reverse cross-linked and purified by extraction with phenol/chloroform, followed by precipitation with ethanol. The DNA concentration of the 3 C libraries was determined using Picogreen fluorescence assay (Invitrogen). 3C-qPCR reactions were performed by the Sybr-Greener Kit (Invitrogen), and the chromatin interactions were normalized by loading control. On the basis of the 4C-seq data, we selected a region in the hTERT coding region with background levels of interaction, with a similar genomic distance from the hTERT bait region as the experimental interaction, to serve as a control interaction. The primers used for 3c-qPCR are listed in Supplementary Table 2.

**Gene expression analysis**. The expression data for multiple cancer types were downloaded from TCGA Firehose (http://gdac.broadinstitute.org/runs/stddata__2016_01_28/data/, Preprocessed Level 3 RSEM log2 normalized). All the relationship of candidate genes was assessed using Spearman's correlation coefficient (r) in samples with non-zero expression. Expression of hTERT was modeled using multiple linear regression (lm()) to determine contribution of individual genes in TGCT and THYM samples. Best linear model was selected using a step wise forward linear regression model. We applied multiple-hypothesis correction by following Benjamini–Hochberg algorithm to calculate false discovery rates (FDRs) based on p values from F statistic of linear regression. The models with FDRs <0.01 were regarded as significant model. All analysis was done in R-3.2.3.

**Neural induction of human embryonic stem cells**. Neural induction of WA01 and WA18 ES cells was carried out using a previously established protocol[40,41]. Briefly, ES cells were cultured in mTeSR1 medium. When the ES cell cultures reached ~20% confluence, mTeSR1 medium was removed and replaced with neural induction media containing DMEM/F12: Neurobasal (1:1), 1xN2, 1xB27, 1% Glutmax, 5 μg/ml BSA, 4 μM CHIR99021 (Cellagentech), 3 μM SB431542 (Cellagentech), 0.1 μM Compound E (γ-Secretase Inhibitor XXI, EMD Chemicals Inc.), 10 ng/ml hLIF (Millipore) for 7 days. The culture was then split 1:3 for the next six passages using Accutase, and the cells were cultured in human neural progenitor cells (NPCs) maintenance media containing DMEM/F12: Neurobasal (1:1), 1xN2, 1xB27, 1% Glutmax, 5 μg/ml BSA, 3 μM CHIR99021, 2 μM SB431542, 10 ng/ml hLIF on matrigel-coated plates. After six passages, the cells were split 1:10 regularly.

**Immunocytochemistry**. For immunofluorescence assays, we have adapted the procedure from previous publications[70]. Briefly, the cells were fixed with 3.7% formaldehyde (vol/vol) in 1× tris-buffered saline (TBS) for 10 min at room temperature. Cells were blocked in permeabilization and blocking buffer (2% normal goat serum, 0.4% trixton X-100, 1 × TBS) for 60 min at room temperature and then incubated with primary antibody, mouse anti-TRF2 (1:50, Merck, #630787) and rabbit anti-γ-H2AX (1:1000, Cell Signaling Technologies, #9718) for 16 hr at 4 degree. The following day, cells were washed with wash buffer (0.4% triton X-100 in 1 × TBS) and incubated with secondary antibody, goat anti-mouse IgG alexa fluor 488 (1:1000, Invitrogen, #A11029) and goat anti-rabbit IgG alexa fluor 568 (1:1000, Invitrogen, #A11036) respectively for 60 min at room temperature. Nuclei were visualized by DAPI staining (Sigma-Aldrich). Images were acquired on IX83 Olympus fluorescence microscope. Images were analyzed using ImageJ software.

**Colony formation assay**. For colony formation assay, 300 cells per condition were plated in triplicates and cultured for two weeks before staining for viable colonies with crystal violet dye.

**Tumor xenograft in NSG mice**. PC-3 cells were suspended in 1× Matrigel at a concentration of $2 \times 10^6$ cells per 100 μl, and the cells were injected subcutaneously into 4- to 6-week-old NSG mice (NOD.Cg-Prkdc$^{scid}$ Il2rg$^{tm1Wjl}$/SzJ) in the dorsolateral area bilaterally. Tumor volume and body weight were measured every week. Tumor volume was calculated using the formula, $S^2 \times L/2$, where $S$ and $L$ represent the small and large diameters of the tumor, respectively. All animal studies were conducted in compliance with animal protocols approved by Institutional Animal Care and Use Committee (IACUC) of Singapore.

**Statistical analysis**. Each assay was repeated in three independent experiments. Statistical significance of differences between groups was analyzed using Student's t test or ANOVA analysis. $P < 0.05$ was considered significant. All the data obtained are presented as mean ± SEM. The relationship of candidate genes was assessed using Spearman's correlation coefficient (r). If the $P$ value is lower than 5% ($P < 0.05$), the correlation coefficient is deemed to be statistically significant.

**Data availability**. ChIP-Seq data are available at Gene Expression Omnibus (GSE97570).

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

## Acknowledgements

We thank Life Science Editors for editing services. This work was supported by grants from MOE tier 2 funding (MOE2014-T2-1-138, Singapore) and NRF competitive research programmme (NRF2016NRF-CRP001-024, Singapore) to S.L.

## Author contributions

T.Y., A.C., D.M., G.M.S., C.X., M.X., L.P., J.W., Y.P.L., J.H.J.H., J.J.Q.N. and M.K.R.: Perform the experiments. W.F.O., A.Q., L.A., S.G.R. and S.G.: Analyzed the data. F.A.B.: Provide guidance and the miRNA inhibitor library for screening. V.T., M.J.F., J.O.J.D. and J.R.H.: Provide technical guidance on 3C technique. P.S., E.G., X.B., M.J.F. and P.T.: Supervised the study. T.Y. and S.L.: Wrote the paper.

## Additional information

**Competing interests:** The authors declare no competing financial interests.

