## [Peer Review File · Nature Communications]

Reviewer #1 (Remarks to the Author):

In this manuscript, the authors identified a novel regulatory mechanism of hTERT expression by HOXC5 and micro RNA - miR615-3p, which is located within intron 1 of the HOXC5 gene. While interesting, several issues need to be clarified.

Major concerns

1. The authors argued in the abstract that “HoxC5 also repressed hTERT expression through disrupting the long-range interaction between hTERT promoter and its distal enhancer”. What is the direct evidence supporting this statement? From the manuscript only correlations are provided. The deactivation system using dCas9_KRAB is not enough to support the conclusions. Also, indicate how those four sgRNAs were introduced in A375 cells and how they changed the epigenetic status.

(a) For example, the EMSA assay using recombinant HoxC5 and the sequences of hTERT upstream enhancer region (-20 kb region), potential HoxC5 binding site (as the authors claimed in Figure S8), would be good enough to support the authors' claims.

(b) At a minimum provide the 3C assays to show the long-range interaction changes.

2. The authors stated that miRNA miR-615-3p is a negative regulator of hTERT expression by targeting the 3' UTR of hTERT gene. The luciferase assay alone is not sufficient to support that argument. The authors should show this by generating the hTERT 3'UTR mutant knock-in in the hTERT locus or at least in an hTERT BAC. For example, a good experiment may be for the authors to mutate the seed sequence in the TERT 3'UTR and then test for TERT expression? One would predict that TERT mRNA levels would increase. This experiment is more direct. The experiment of mutating the miRNA could have vast indirect effects and thus could both directly and indirectly effect TERT. The authors should consider mutating the endogenous TERT miRNA seed sequence and measuring TERT levels. Also consider replacing the entire TERT 3' UTR with a different sequence to avoid potential problems,

3. The telomere elongation upon HOXC5 is not strong and it may be that the telomeres are not elongating in the Figure 3F experiment. How many cell divisions after shRNA KD of HOXC5 did the authors wait before measuring telomere length in the HeLa clones? This information is vital for the interpretation of the results. The telomere shortening (Figure 3D) also needs to have the number of cell replications (population doublings) and a number of lanes at different times, otherwise this could be solely due to clonal variability in telomere length. In each figure and in the text a series of different time points (population doublings should be shown). For example, on Figure 2K how long after microRNA KO (KO-1 and KO-2) are the TRFs shown. Do the investigators has several different time points? Do the telomeres continue to elongate or reach a new steady state? It is important to detail how much telomeres grow in a specific time period. In addition, it would be helpful to show a TRAP gel and not just Q-PCR TRAP.

4. TERT repression does not result in smaller tumors unless telomere length is limiting, which is doubtful in these xenografts shown in the paper. Clearly miR 615-3p and HOXC5 are regulating other genes related to growth and differentiation, this should be mentioned in the xenograft section. If the authors think telomere length is limiting then this should be stated and evidence provided. HOXC5 has a much larger effect (as the authors mention) on both telomere length (Figure 3D) and xenograft growth in PC-3 cells compared to others. However, the differences in growth curves is not that substantial. If the investigators passaged the cells longer in vitro and retested tumor growth it may eventually be more dramatic as a proof of principal.

Minor concerns

1. The pull down data to show that these proteins are involved in repression/activation of TERT transcription is distracting where it is presented (although interesting). Consider moving the differentiation studies up before focusing on all the protein interactions. This would make the paper flow better.
2. Why would a neuroendocrine type cancers be more sensitive compared to other tumor types?
3. page 4 line 69-70: hTERC, not hTER
4. page 10 line 234-235 “a positive correlation between the expression of miR-615-3p and HOXC5 mRNA (data not shown)”: Provide the data or revise this statement.
5. Figure 2 H-K: What is the expression level of HoxC5 in KO clones?
6. Importantly, each figure and subfigures should clearly identify what cells are being tested without having to read the legend or to go back to the text. For example, on Figure 7 it is unclear what cells these are.

Reviewer #2 (Remarks to the Author):

In this manuscript titled ‘HoxC5 and miRNA-615-3p target newly evolved genomic regions to repress hTERT and inhibit tumorigenesis’ authors attempt to understand how telomerase activity is repressed during human development and disease. More specifically, the authors appear to be interested to understand how matured tissue cells upon primitive cell differentiation repress telomerase activity and by doing so in malignant cells, raise the threshold for malignant clones to propagate indefinitely. With the help of genome-wide miRNA inhibitor library and biochemical experiments, the authors screen for novel regulators of hTERT UTR regions and come to the conclusion that human intronic microRNA mir-615-3p may ‘fine-tune’ stability of hTERT mRNA, and more interestingly claim that HoxC5 gene, which harbors mir-615-3p in its intron-1, may

developmentally-control synthesis of hTERT gene via regulating a 20kb upstream enhance element. In summary, HoxC5 and mir-615-3p join a growing list of molecules described as negative regulators of hTERT expression and its telomeric function. The statistical tests are properly conducted.

Concerns:

Related to evolutionary link:

-The claim is that HoxC5 and its intronic mir-615 are conserved in placental mammals (mice and human) but the action on hTERT is restricted to long-lived mammals (human) due to the evolution of novel genomic regions. Is the DNA sequence within -20kb TERT enhancer region, suggested to be a putative target for HoxC5 complex also unique to long-lived old-world monkeys (human)? If yes then please provide evidence and if no then the manuscript title is not justified. Other old-world monkeys whose genomic sequences are available should be included in phylogenetic analyses to strengthen the notion that the biology described here is primarily a product of recent evolution within mammalian kingdom.

-The developmental advantage in species (human) that carry HoxC5/mir-615 regulated hTERT is not clear since the data presented is solely using laboratory models of one species. The manuscript suggests that long-lived monkey species has 'gained' additional control mechanisms to prevent hTERT expression 'leakage' thereby controlling hTERT more robustly during development. The study would be strengthened if experiments were performed using functionally equivalent cells from mammalian species that lack this developmental advantage.

-The study tries to link the 'newly-evolved' genomic regulation at hTERT to cellular behaviors like reprogramming and cancer cell growth. Since no genetic system has evolved, to my knowledge, to cope with 'reprogramming' or developmental-fate of 'fully' malignant cancer cells, therefore the artificial cell culture systems, which are difficult to reproduce, fail to capture the utility of this mechanism in real world and therefore is the weakest link in this otherwise interesting study.

Related to MicroRNA-615:

-The microRNA inhibitor screen, hTERT UTR reporter assays, telomerase activity and telomere length studies using microRNA knockout experiments conclusively point towards significant but 'minor' contribution of this microRNA in cancer cell-type specific manner and certainly not 'dramatic' effects on hTERT mRNA levels and telomerase activity in HeLa cells as claimed in page-8 line 184.

-Are there transcription factor targets of mir-615 which could also contribute to changes in hTERT expression and telomere length?

-Does the telomere length progressively increase in mir-615 knockout cell lines or do they reach a maximum telomere length?

-In line 476, it is mentioned that "... we have shown that the hTERT 5'UTR AND 3'UTR play important roles in regulation of hTERT..". This study does not suggest any role for 5'UTR. Please correct.

Related to HoxC5:

-Is HoxC5 ubiquitously expressed in non-stem cells in human tissues? The authors make an argument in favor of certain species specific function of HoxC5 on telomeric transcriptional control but not a single normal tissue is presented in credible manner which may take 'advantage' of the HoxC5 mediated regulation of hTERT?

-The data that tries to link hTERT and HOXC5 in different human cancers is weak and speculative.

Reviewer #1

1. *The authors argued in the abstract that “HoxC5 also repressed hTERT expression through disrupting the long-range interaction between hTERT promoter and its distal enhancer”. What is the direct evidence supporting this statement? From the manuscript only correlations are provided. The deactivation system using dCas9_KRAB is not enough to support the conclusions. Also, indicate how those four sgRNAs were introduced in A375 cells and how they changed the epigenetic status.*

(a) For example, the EMSA assay using recombinant HoxC5 and the sequences of hTERT upstream enhancer region (-20 kb region), potential HoxC5 binding site (as the authors claimed in Figure S8), would be good enough to support the authors' claims.

(b) At a minimum provide the 3C assays to show the long-range interaction changes.

>>> Based on the reviewer's request, we have performed 3C qPCR in PC-3 cells overexpressing GFP or HOXC5. As shown in Supplementary Figure 10, our results indicated that overexpression of HOXC5 in PC-3 cells resulted in reduced interaction between hTERT promoter and its upstream enhancer specifically. These results further support our hypothesis that HoxC5 represses hTERT expression by disrupting the long-range interaction between hTERT promoter and its distal enhancer.

For the CRISPRi experiment, single vector co-expressing 4 sgRNAs¹ and Neomycin cassette was transiently transfected into A375 cells stably transduced with pLV-hUbc-dCas9-KRAB-T2A-Puro. The cells with the sgRNA-expressing vector were enriched using G418. The pooled cells were used for ChIP-PCR. As shown in Supplementary Figure 9C, targeting of dCas9-KRAB fusion protein to the hTERT upstream enhancer region resulted in a corresponding depletion of RNA Pol II as well as permissive H3K4me1 and H3K27ac marks from hTERT promoter and enhancer regions.

2. *The authors stated that miRNA miR-615-3p is a negative regulator of hTERT expression by targeting the 3' UTR of hTERT gene. The luciferase assay alone is not sufficient to support that argument. The authors should show this by generating the hTERT 3'UTR mutant knock-in in the hTERT locus or at least in an hTERT BAC. For example, a good experiment may be for the authors to mutate the seed sequence in the TERT 3'UTR and then test for TERT expression? One would predict that TERT mRNA levels would increase. This experiment is more direct. The experiment of mutating the miRNA could have vast indirect effects and thus could both directly and indirectly effect TERT. The authors should consider mutating the endogenous TERT miRNA seed sequence and measuring TERT levels. Also consider replacing the entire TERT 3' UTR with a different sequence to avoid potential problems,*

>>> We thank the reviewer for the suggestion. We have engineered small deletion as well as knock-in mutation in the *hTERT* 3'UTR in RKO cells using CRISPR-mediated gene targeting². As shown in Supplementary Figure 5, bi-allelic deletion or mutation of the seed region in the *hTERT* 3'UTR resulted in an increase in *hTERT* expression. In contrast, small deletion right next to the seed region does not affect the expression of endogenous *hTERT*.

3. The telomere elongation upon *HOXC5* is not strong and it may be that the telomeres are not elongating in the Figure 3F experiment. How many cell divisions after shRNA KD of *HOXC5* did the authors wait before measuring telomere length in the HeLa clones? This information is vital for the interpretation of the results. The telomere shortening (Figure 3D) also needs to have the number of cell replications (population doublings) and a number of lanes at different times, otherwise this could be solely due to clonal variability in telomere length. In each figure and in the text a series of different time points (population doublings should be shown). For example, on Figure 2K how long after microRNA KO (KO-1 and KO-2) are the TRFs shown. Do the investigators has several different time points? Do the telomeres continue to elongate or reach a new steady state? It is important to detail how much telomeres grow in a specific time period. In addition, it would be helpful to show a TRAP gel and not just Q-PCR TRAP.

>>> We thank the reviewer for this comment. We have re-labeled the Figures with teloblot results to provide the number of population doublings in each lane. We also included telomere length measurement from different population doublings for each cell line. For *HOXC5* overexpression and shRNA knockdown, pooled cells transduced with the lentivirus were enriched using antibiotic selection to avoid the potential problem of clonal variability. As shown in Figure 2 and 3, the telomere length reaches a steady state after extended culture *in vitro*. We also provide TRAP gel instead of Q-PCR TRAP in Figure 2 and Figure 3 as suggested by the reviewer.

4. TERT repression does not result in smaller tumors unless telomere length is limiting, which is doubtful in these xenografts shown in the paper. Clearly miR 615-3p and HOXC5 are regulating other genes related to growth and differentiation, this should be mentioned in the xenograft section. If the authors think telomere length is limiting then this should be stated and evidence provided. HOXC5 has a much larger effect (as the authors mention) on both telomere length (Figure 3D) and xenograft growth in PC-3 cells compared to others. However, the differences in growth curves is not that substantial. If the investigators passaged the cells longer in vitro and retested tumor growth it may eventually be more dramatic as a proof of principal.

>>> We agree with the reviewer that miR-615-3p and HoxC5 regulate other genes related to cell growth and differentiation as well. As shown in Supplementary Table S1, at least 59 potential targets of miR-615-3p have been predicted. Many of these candidates, such as *PTK2B*, *MEF2A* and *DACF6* are known to regulate cell growth and differentiation. In addition, our ChIP-seq data indicate that HoxC5 preferentially target genes in the TGF- β pathway, which plays well-characterized roles in cell growth and differentiation³. As shown in Supplementary Figure 13, overexpression of *HOXC5* indeed affected the expression of genes in the TGF- β pathway.

In addition, overexpression of *HOXC5* in PC-3 cells resulted in dramatic telomere shortening (Figure 3D). The telomere shortening in the PC-3 cells overexpressing *HOXC5* led to telomere deprotection as increased telomere dysfunction-induced foci were detected (Supplementary Figure 15). This may contribute to cell growth inhibition in PC-3 cells overexpressing *HOXC5*.

Previous studies also point to the possibility of telomere-independent function of hTert in promoting cancer cell proliferation and survival^{4,5}. Repression of *hTERT* expression by *HOXC5* overexpression may inhibit cancer cell proliferation through telomere-independent role of hTert as well.

We choose not to passage cells overexpressing *HOXC5* for too long as cells overexpressing *HOXC5* is expected to have a negative selection pressure to survive. We expect the cells that survive long-term culture *in vitro* to have low expression of *HOXC5*, and therefore less dramatic cell growth inhibition effect both *in vitro* and *in vivo*.

Minor concerns

1. *The pull down data to show that these proteins are involved in repression/activation of TERT transcription is distracting where it is presented (although interesting). Consider moving the differentiation studies up before focusing on all the protein interactions. This would make the paper flow better.*

>>> We have amended the text as suggested.

2. *Why would a neuroendocrine type cancers be more sensitive compared to other tumor types?*

>>> We thank the reviewer for this comment. We are not emphasizing that only PC-3 cells are sensitive to *HOXC5* overexpression. Using CellTiter-Glo luminescent cell viability assay, we observed inhibition of cell proliferation in PC-3, HeLa and BT549 cells that overexpress *HOXC5* (Supplementary Figure 14D-14F).

3. *page 4 line 69-70: hTERC, not hTER*

>>> We have amended the text as suggested.

4. *page 10 line 234-235 “a positive correlation between the expression of miR-615-3p and HOXC5 mRNA (data not shown)”: Provide the data or revise this statement.*

>>> We have deleted this statement. We have instead reiterated this point later on using RNA-seq data from TCGT and THYM cancer samples (Figure 7H and 7K).

5. *Figure 2 H-K: What is the expression level of HoxC5 in KO clones?*

>>> We used single sgRNA to generate clonal derived RKO cell lines with small deletion in the hairpin region of miR-615 located in intron 1 of *HOXC5*. As shown in the Supplementary Figure 5C and 5D, there is no detectable change in *HOXC5* mRNA or HoxC5 protein expression level in *mir-615-3p* knockout clones.

6. Importantly, each figure and subfigures should clearly identify what cells are being tested without having to read the legend or to go back to the text. For example, on Figure 7 it is unclear what cells these are.

>>> We thank the reviewer for this comment. We have relabeled our Figures to indicate the cell lines used in each Figure.

Reviewer #2:

1. The claim is that HoxC5 and its intronic mir-615 are conserved in placental mammals (mice and human) but the action on hTERT is restricted to long-lived mammals (human) due to the evolution of novel genomic regions. Is the DNA sequence within -20kb TERT enhancer region, suggested to be a putative target for HoxC5 complex also unique to long-lived old-world monkeys (human)? If yes then please provide evidence and if no then the manuscript title is not justified. Other old-world monkeys whose genomic sequences are available should be included in phylogenetic analyses to strengthen the notion that the biology described here is primarily a product of recent evolution within mammalian kingdom.

>>> We thank the reviewer for this comment. Based on the available genomic sequence from UCSC genome database (Vertebrate Multiz Alignment & Conservation-100 species), we have compared the sequence of *TERT* genomic loci across many species including old-world monkeys (Supplementary Figure 11). As highlighted in the sequence alignment, the -20kb *TERT* enhancer region is indeed conserved in the long-lived old-world monkey and human. However, this region is not conserved in the remaining >80 species of short-lived mammals and other vertebrates (including mouse, rat, dog, cow and zebrafish).

2. The developmental advantage in species (human) that carry HoxC5/mir-615 regulated hTERT is not clear since the data presented is solely using laboratory models of one species. The manuscript suggests that long-lived monkeys species has 'gained' additional control mechanisms to prevent hTERT expression 'leakage' thereby controlling hTERT more robustly during development. The study would be strengthened if experiments were performed using functionally equivalent cells from mammalian species that lack this developmental advantage.

>>> We thank the reviewer for this comment. We have overexpressed miR-615-3p (human and mouse miR-615-3p are identical) and mouse *HOXC5* in three mouse cell lines: B16, Hepa 1-6 and Renca. As shown in Supplementary Figure 12C-12N, overexpression of miR-615-3p or mouse *HOXC5* did not suppress the expression of mouse *TERT*.

3. The study tries to link the 'newly-evolved' genomic regulation at hTERT to cellular

behaviors like reprogramming and cancer cell growth. Since no genetic system has evolved, to my knowledge, to cope with ‘reprogramming’ or developmental-fate of ‘fully’ malignant cancer cells, therefore the artificial cell culture systems, which are difficult to reproduce, fail to capture the utility of this mechanism in real world and therefore is the weakest link in this otherwise interesting study.

>>> We agree with the reviewer that no genetic system has evolved to cope with developmental-fate of “fully” malignant cancer cells. Telomerase activity is repressed in most human somatic tissues during development⁶⁻¹¹. The lack of telomerase activity in somatic cells results in progressive telomere shortening in subsequent cell division. When telomere become critically short, they trigger DNA damage response and induce cell growth arrest, known as replicative aging, which provides a barrier for tumor initiation and progression^{12,13}. Telomerase activation is one critical step in tumorigenesis and about 85-90% of all human cancers rely on telomerase activity for telomere maintenance. There are substantial genetic evidences supporting the theory that telomere length co-evolves with lifespan¹⁴. Previous studies from labs of Dr. Woody Wright and Dr. Jerry Shay have examined telomere length and telomerase expression in cultured fibroblast from >60 mammalian species. Their results indicate that telomere length is inversely correlated with lifespan, and telomerase expression co-evolved with body size¹⁴. The repressed telomerase activity and short telomere length (as observed in human) limit the proliferative capacity of somatic cells, and establish the basis for replicative aging that is critical in suppressing tumor formation over long period of time in long-lived mammals. However, this program is often not conserved in short-lived species, such as mouse. It is hypothesized that the high telomerase activity and long telomere length in short-lived mammals may be a trade-off for reducing the energetic/cellular costs of oxidative protection mechanisms.

Previous studies using transgenic mouse lines with bacterial artificial chromosomes have uncovered that the *cis*-acting regulatory elements in the *hTERT* genomic locus are pivotal in mediating its silencing during normal development¹⁵⁻¹⁸. Our data provide the evidence that *hTERT* 3’UTR and upstream enhancer region are these vital *cis*-regulatory elements responsible for repressing *hTERT* expression upon cellular differentiation.

4. The microRNA inhibitor screen, hTERT UTR reporter assays, telomerase activity and telomere length studies using microRNA knockout experiments conclusively point towards significant but ‘minor’ contribution of this microRNA in cancer cell-type specific manner and certainly not ‘dramatic’ effects on hTERT mRNA levels and telomerase activity in HeLa cells as claimed in page-8 line 184.

Are there transcription factor targets of mir-615 which could also contribute to changes in hTERT expression and telomere length?

>>> We have searched the potential targets of miR-615-3p using the microT-CDS algorithm¹⁹. A total of 59 potential targets have been predicted (Supplementary Table S1). Among these 59 potential targets, 56 are also predicted by Targetscan²⁰. One of the target-RNASEK has been experimentally verified. Based on current literature, none of the 59 targets is directly involved in regulation of *hTERT* expression or telomere length.

Based on our results, inhibition of miR-615-3p using the hairpin inhibitor resulted in 1.5 to 2 fold increase in *hTERT* expression. For comparison, *hTERT* promoter mutations have been

shown to increase *hTERT* promoter transcriptional activity by two- to four-fold²¹⁻²³, which is sufficient to promote tumorigenesis.

5. *Does the telomere length progressively increase in mir-615 knockout cell lines or do they reach a maximum telomere length?*

>>> The miR-615-3p knockout cell lines are derived from single cell. By the time we have the clonal cell line, they have undergone at least 20 population doublings (PD20). As shown in the Figure 2K, we did not observe further elongation of telomeres at PD30 compared to PD20.

6. *In line 476, it is mentioned that "... we have shown that the hTERT 5'UTR AND 3'UTR play important roles in regulation of hTERT..". This study does not suggest any role for 5'UTR. Please correct.*

>>> We have amended the text as suggested.

7. *Is HoxC5 ubiquitously expressed in non-stem cells in human tissues? The authors make an argument in favor of certain species specific function of HoxC5 on telomeric transcriptional control but not a single normal tissue is presented in credible manner which may take 'advantage' of the HoxC5 mediated regulation of hTERT?*

>>> There is very limited literature on expression and function of human HOXC5. Based on the RNA-seq data from "The human Protein Atlas", *HOXC5* is expressed in many human tissues (Supplementary Figure 16) and cell types (<http://www.proteinatlas.org/ENSG00000172789-HOXC5/tissue/primary+data>). For example, high expression of HOXC5 can be detected in cells from seminiferous ducts in the testis.

In addition, our results in Figure 5 have shown the dramatic up-regulation of *HOXC5* and miR-615-3p expression upon cellular differentiation of normal human embryonic stem (ES) cells. These results are consistent with our finding that HoxC5 and miR-615-3p negatively regulate *hTERT* expression.

Consistent with our observations, previous studies have shown that *HOXC5* is involved in lymphomagenesis. Strong expression of *HOXC5* can be detected in cells representing mature stages of lymphoid cell differentiation, while no *HOXC5* expression can be detected in leukemic cells representing immature stages of lymphoid differentiation²⁴. These results suggest a possible role of *HOXC5* in suppressing tumorigenesis.

8. *The data that tries to link hTERT and HOXC5 in different human cancers is weak and speculative.*

>>> We agree with the reviewer that correlation does not mean causation. Therefore we have changed our wording in the manuscript to explain this better. As indicated in the manuscript, we observed negative correlation between *HOXC5* and *hTERT* expression

across multiple cancer types, the highest being TGCT and THYM. Both cancers originate from telomerase-rich tissues that rarely harbor *hTERT* promoter mutations^{25,26}. This is not likely to be random: while the expression of *HOXC5* and miR-615 is co-regulated (Figure 7H and 7K), we did not see significant negative correlation between the expression of miR-615-3p and *hTERT* in TGCT and THYM (Figure 7G and 7J). This is consistent with *HoxC5* playing a more important role in regulating *hTERT* expression as we have shown in the manuscript. Whether the loss of *HOXC5* expression in these cancers indeed contributes to the up-regulation of *hTERT* remains to be addressed, but this is outside the scope of this manuscript.

References:

- 1 Sakuma, T., Nishikawa, A., Kume, S., Chayama, K. & Yamamoto, T. Multiplex genome engineering in human cells using all-in-one CRISPR/Cas9 vector system. *Sci Rep* **4**, 5400 (2014).
- 2 Ran, F. A. *et al.* Genome engineering using the CRISPR-Cas9 system. *Nat Protoc* **8**, 2281-2308 (2013).
- 3 Massague, J. TGFbeta signalling in context. *Nat Rev Mol Cell Biol* **13**, 616-630 (2012).
- 4 Ye, J., Renault, V. M., Jamet, K. & Gilson, E. Transcriptional outcome of telomere signalling. *Nat Rev Genet* **15**, 491-503 (2014).
- 5 Li, Y. & Tergaonkar, V. Noncanonical functions of telomerase: implications in telomerase-targeted cancer therapies. *Cancer Res* **74**, 1639-1644 (2014).
- 6 Kim, N. W. *et al.* Specific association of human telomerase activity with immortal cells and cancer. *Science* **266**, 2011-2015 (1994).
- 7 Wright, W. E., Piatyszek, M. A., Rainey, W. E., Byrd, W. & Shay, J. W. Telomerase activity in human germline and embryonic tissues and cells. *Dev Genet* **18**, 173-179. (1996).
- 8 Feng, J. *et al.* The RNA component of human telomerase. *Science* **269**, 1236-1241. (1995).
- 9 Nakamura, T. M. *et al.* Telomerase catalytic subunit homologs from fission yeast and human. *Science* **277**, 955-959. (1997).
- 10 Bernardes de Jesus, B. & Blasco, M. A. Telomerase at the intersection of cancer and aging. *Trends Genet* **29**, 513-520 (2013).
- 11 Collins, K. & Mitchell, J. R. Telomerase in the human organism. *Oncogene* **21**, 564-579 (2002).
- 12 Armitage, P. & Doll, R. The age distribution of cancer and a multi-stage theory of carcinogenesis. *Br J Cancer* **8**, 1-12 (1954).
- 13 Luebeck, E. G. & Moolgavkar, S. H. Multistage carcinogenesis and the incidence of colorectal cancer. *Proc Natl Acad Sci USA* **99**, 15095-15100 (2002).
- 14 Gomes, N. M. *et al.* Comparative biology of mammalian telomeres: hypotheses on ancestral states and the roles of telomeres in longevity determination. *Aging Cell* **10**, 761-768 (2011).
- 15 Wang, S., Hu, C. & Zhu, J. Transcriptional silencing of a novel hTERT reporter locus during in vitro differentiation of mouse embryonic stem cells. *Mol Biol Cell* **18**, 669-677 (2007).
- 16 Jia, W. *et al.* A BAC transgenic reporter recapitulates in vivo regulation of human telomerase reverse transcriptase in development and tumorigenesis. *Faseb J* **25**, 979-989 (2011).

- 17 Cheng *et al.* Repression of telomerase gene promoter requires human-specific genomic context and is mediated by multiple HDAC1-containing corepressor complexes. *Faseb J* (2016).
- 18 Forsyth, N. R., Wright, W. E. & Shay, J. W. Telomerase and differentiation in multicellular organisms: turn it off, turn it on, and turn it off again. *Differentiation* **69**, 188-197 (2002).
- 19 Paraskevopoulou, M. D. *et al.* DIANA-microT web server v5.0: service integration into miRNA functional analysis workflows. *Nucleic Acids Res* **41**, W169-173 (2013).
- 20 Lewis, B. P., Burge, C. B. & Bartel, D. P. Conserved seed pairing, often flanked by adenosines, indicates that thousands of human genes are microRNA targets. *Cell* **120**, 15-20 (2005).
- 21 Horn, S. *et al.* TERT promoter mutations in familial and sporadic melanoma. *Science* **339**, 959-961 (2013).
- 22 Huang, F. W. *et al.* Highly recurrent TERT promoter mutations in human melanoma. *Science* **339**, 957-959 (2013).
- 23 Killela, P. J. *et al.* TERT promoter mutations occur frequently in gliomas and a subset of tumors derived from cells with low rates of self-renewal. *Proc Natl Acad Sci USA* **110**, 6021-6026 (2013).
- 24 Bijl, J. *et al.* Expression of HOXC4, HOXC5, and HOXC6 in human lymphoid cell lines, leukemias, and benign and malignant lymphoid tissue. *Blood* **87**, 1737-1745 (1996).
- 25 Carcano, F. M. *et al.* Hotspot TERT promoter mutations are rare events in testicular germ cell tumors. *Tumour Biol* **37**, 4901-4907 (2016).
- 26 Huang, D. S. *et al.* Recurrent TERT promoter mutations identified in a large-scale study of multiple tumour types are associated with increased TERT expression and telomerase activation. *Eur J Cancer* **51**, 969-976 (2015).

Reviewer #1 (Remarks to the Author):

the authors have adequately responded to the original concerns.

Reviewer #2 (Remarks to the Author):

The revised manuscript clearly explains how mir-615-3p and its host gene HOXC5 both repress hTERT by disrupting its promoter interaction with its distal enhancer, regions argued by the authors to be conserved and unique to species closely linked to humans and not relevant in common laboratory models of cancer studies including mice and rats. Authors also provide convincing evidence as to how HOXC5 and miR-615-3p in human cancer cells inhibits hTERT expression and suppresses cancer cell growth.

I am overall satisfied with the reviewers response. Shang Li's group provide insight into a brand new molecular mechanism which will be well received by your readers. I recommend this manuscript for publication in its entirety.

Dear Dr. Stephanie Koo,

Thank you very much for reviewing our manuscript “HoxC5 and miR-615-3p target newly evolved genomic regions to repress *hTERT* and inhibit tumorigenesis”. I am glad that the paper is in the final step before being accepted for publication in Nature Communications. Please find our point-by-point response to reviewer comments below (>>>):

Reviewer #1

1. The authors have adequately responded to the original concerns.

>>> We are glad that reviewer #1 is satisfied with our revised manuscript.

Reviewer #2:

The revised manuscript clearly explains how mir-615-3p and its host gene HOXC5 both repress hTERT by disrupting its promoter interaction with its distal enhancer, regions argued by the authors to be conserved and unique to species closely linked to humans and not relevant in common laboratory models of cancer studies including mice and rats. Authors also provide convincing evidence as to how HOXC5 and miR-615-3p in human cancer cells inhibits hTERT expression and suppresses cancer cell growth.

I am overall satisfied with the reviewers response. Shang Li's group provide insight into a brand new molecular mechanism which will be well received by your readers. I recommend this manuscript for publication in its entirety.

>>> We are glad that reviewer #2 is satisfied with our responses to his concerns.